# High-efficiency reinforcement learning with hybrid architecture photonic integrated circuit

Xuan-Kun Li [1,2], Jian-Xu Ma[3], Xiang-Yu Li[3], Jun-Jie Hu[1,2], Chuan-Yang Ding[1,2], Feng-Kai Han[1,2], Xiao-Min Guo[3], Xi Tan [1,2] & Xian-Min Jin [1,2,3,4] ✉

Reinforcement learning (RL) stands as one of the three fundamental paradigms within machine learning and has made a substantial leap to build general-purpose learning systems. However, using traditional electrical computers to simulate agent-environment interactions in RL models consumes tremendous computing resources, posing a significant challenge to the efficiency of RL. Here, we propose a universal framework that utilizes a photonic integrated circuit (PIC) to simulate the interactions in RL for improving the algorithm efficiency. High parallelism and precision on-chip optical interaction calculations are implemented with the assistance of link calibration in the hybrid architecture PIC. By introducing similarity information into the reward function of the RL model, PIC-RL successfully accomplishes perovskite materials synthesis task within a 3472-dimensional state space, resulting in a notable 56% improvement in efficiency. Our results validate the effectiveness of simulating RL algorithm interactions on the PIC platform, highlighting its potential to boost computing power in large-scale and sophisticated RL tasks.

Machine Learning (ML) within Artificial Intelligence (AI) brings revolutionary transformations across nearly all industries[1–5]. Reinforcement learning (RL)[6], one of the three basic ML paradigms alongside supervised and unsupervised learning, is becoming a remarkably attractive ML approach, spanning applications from strategy games[7] to robotics[8,9] and autonomous control[10,11]. As the first computer program to defeat a professional human Go player, AlphaGo operates on RL principles[12,13]. Additionally, reinforcement learning from human feedback (RLHF)[14] plays a crucial role in enhancing generative pre-trained transformer (GPT) by incorporating valuable insights and knowledge provided by human feedback[15]. RL focuses on the interaction between "agent" and "environment", seeking to derive an optimal policy through the training process. Off-policy RL can learn from large, previously collected datasets, which increases the efficiency of resource utilization and minimizes resource consumption in interactions. One of the most well-known off-policy RL strategies is Q-learning[16], which aims to determine the optimal policy by maximizing the expected value of the total reward across all successive steps.

Taking advantage of the intrinsic high parallelism and bandwidth of photons, combined with highly compact and phase-stable optoelectronic integrated technology, integrated optical computing, encompassing optical neural network (ONN)[17–27], optical quantum computing[28–32] and NP problem solving[33], has not only captured significant interest within academia but also gained widespread recognition within the industry. In recent years, integrated optical computing has shown the potential to achieve state-of-the-art computing power and energy efficiency. This novel computing architecture is anticipated to maintain the pace of Moore's Law[34]. Previous research has predominantly shown the success of combining AI algorithms with ONN in supervised learning tasks, including

[1]Center for Integrated Quantum Information Technologies (IQIT), School of Physics and Astronomy and State Key Laboratory of Advanced Optical Communication Systems and Networks, Shanghai Jiao Tong University, Shanghai 200240, China. [2]Hefei National Laboratory, Hefei 230088, China. [3]TuringQ Co., Ltd., Shanghai 200240, China. [4]Chip Hub for Integrated Photonics Xplore (CHIPX), Shanghai Jiao Tong University, Wuxi 214000, China. ✉e-mail: xianmin.jin@sjtu.edu.cn

classification[17–21,23] and regression[35]. However, the infrequent application of RL in PIC[36] emphasizes the necessity to expand the scope of AI applications within integrated optical computing. Furthermore, the progress of integrated optical computing is impeded by inherent limitations in single architectures, such as Mach-Zehnder interferometers (MZI) mesh[37,38] and coherent linear architectures[39], which include restricted scalability and functionality.

In this work, we experimentally demonstrate the improvements in RL efficiency by using the PIC platform to implement agent-environment interactions. We design a hybrid architecture PIC (HyArch PIC) with remarkable scalability and versatile functionality compared to single integrated optical computing architectures. Co-integrating HyArch PIC with high-speed FPGA and electrical drivers on a single development board results in a highly integrated optoelectronic computing board with a vast optimization space. Through global parameter optimization and link calibration, HyArch PIC exhibits the capability to perform optical dot product operations in dimensions up to 15, ensuring the execution of the subsequent RL algorithm on the PIC. The introduction of similarity information into the reward function, termed similarity reward function (SRF) RL, leads to an exponential acceleration over constant reward function (CRF) RL in the cliff walking benchmark. Additionally, we calculate the similarity of 3472 14-dimensional atom vectors and leverage PIC-RL for the perovskite materials synthesis task, achieving an impressive 56% efficiency improvement. Notably, the highly scalable HyArch PIC shows promising potential in outperforming existing electronic computing architectures in computing power performance, thereby significantly advancing the development of next-generation RL.

## Results

The schematic of HyArch PIC is shown in Fig. 1a, comprising a unitary MZI mesh module for routing and weight distribution, along with three OCTOPUS (Optical CompuTing Of dot-Product UnitS)[39] modules for dot product and matrix-vector multiplication (MVM) calculations. All four modules are integrated on a single chip, ensuring both stability and reconfigurability for advanced photonic computing. Fundamentally, our HyArch PIC possesses the capability to execute high-precision arbitrary real number dot-product operations up to 15 dimensions. The concept of PIC-assisted reinforcement learning (PIC-RL) is illustrated in Fig. 1b. Preprocessed state and action information is encoded, either in amplitude or phase, into the PIC. Subsequently, the PIC simulates the agent-environment interaction using the encoded action and state information. The resulting output light carries reward information for the current state-action pair, contributing to the construction of the reward table (R-table). Through RL training with the PIC R-table, Q values are derived and organized into a tabular format known as the Q-table. Since Q-learning is an off-policy value-based RL method, the well-trained Q-table guides the agent in exploiting the environment by selecting the optimal action, ultimately establishing the optimal policy.

### Hybrid architecture photonic integrated circuit

The top-level diagram of our optoelectronic computing system is illustrated in Fig. 2a. The HyArch PIC (Fig. 2c) and a multi-channel FPGA are co-integrated on a single development board, enabling communication with a computer via a LAN port. A standard server rack

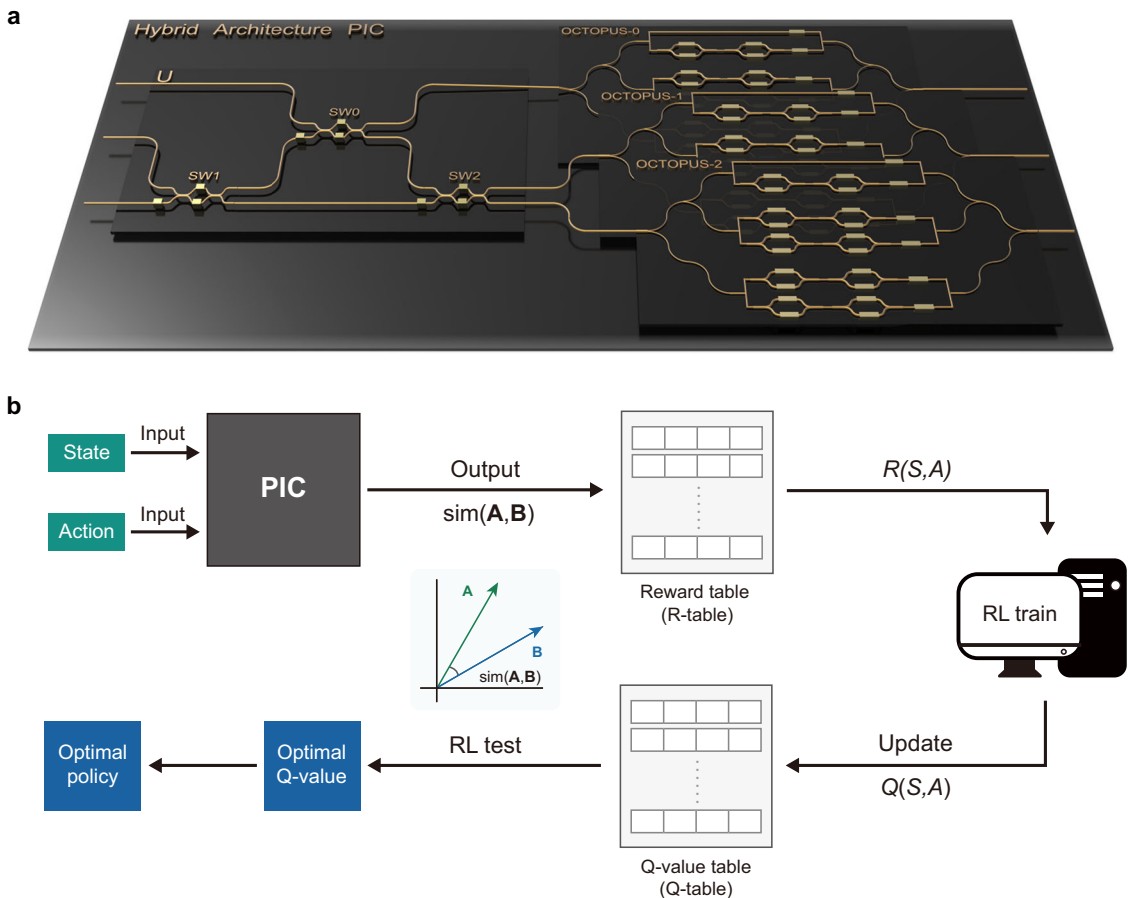

**Fig. 1 | HyArch PIC and PIC-RL concepts. a** Schematic of the proposed hybrid architecture PIC (HyArch PIC), comprising a unitary MZI mesh module and three identical parallel OCTOPUS modules. **b** PIC-assisted reinforcement learning (PIC-RL) leverages PIC for efficient simulation of the agent-environment interaction in the RL algorithm.

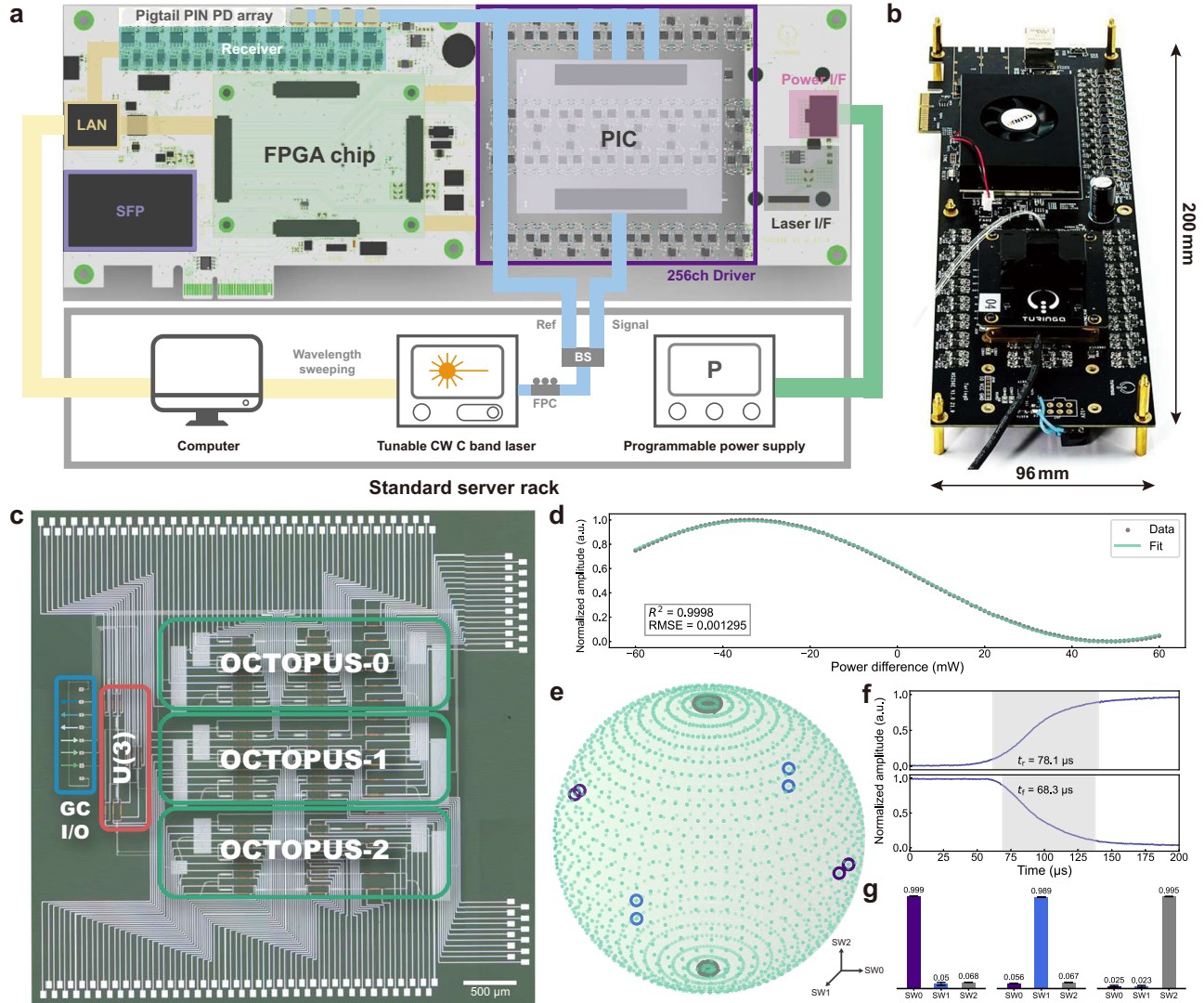

**Fig. 2 | Experimental demonstration of HyArch PIC optoelectronic computing system. a** Top-level diagram of the optoelectronic computing system with integrated PIC and FPGA on a development board. The computer, laser, and power supply are housed in a standard server rack. **b** Photograph of the optoelectronic computing board with the size of 200 mm × 96 mm is shown in Fig. 2b. **c** Microscope image of the HyArch PIC, featuring a grating coupler (GC)-based I/O port array. Three input ports connect to the unitary MZI mesh module, and three output ports export light from the OCTOPUS modules. **d** Unit calibration curve of a single push-pull MZI unit with sine-like fitting. **e** Measurement results for arbitrarily configurable U(3) module. **f** Rising and falling edge of the thermal optical modulator. **g** Power distribution at vertices of 3D spherical coordinate axes, with error bars representing the standard deviations within each group of vertices.

accommodates the control computer, optical signal laser, and programmable power supply. The photograph of the optoelectronic computing board with the size of 200 mm × 96 mm is shown in Fig. 2b. Advanced integration in the optoelectronic computing system enables Python programming for multi-channel modulation and input wavelength sweeping, facilitating HyArch PIC optimization and reconfiguration. Optimized by the simulated annealing algorithm, the overall on-chip loss of 6.5 dB highlights the maturity of the PIC design and manufacturing (see "Methods" and Supplementary Section 1). Figure 2d displays the calibration curve for a single push-pull MZI unit, obtained by sweeping the modulation power difference between the upper and lower arms. This well-fitted curve to the sine-like function $y = a \cdot \sin(bx + c) + d$ ($R^2 = 0.9998$, RMSE = 0.001295) ensures precise encoding and system phase stability. Based on the unit calibration, the U(3) module empowers the flexible configuration of input optical power for the three OCTOPUS modules. Sweeping the three switching units (SW0/SW1/SW2) within the U(3) module maps the normalized output intensity of U(3) to a spherical surface in three-dimensional

space (Fig. 2e). The data points evenly cover the entire 3D spherical surface, demonstrating the U module's ability to achieve arbitrary U(3) transformations. Bar plots in Fig. 2g depict the data near the axis points (marked by circles on the 3D sphere), revealing a high switch extinction ratio. The response time of the thermal optical modulator is measured by an arbitrary waveform generator and oscilloscope, as shown in Fig. 2f, with rising time $t_r$ of 78.1 μs and falling time $t_f$ of 68.3 μs, corresponding to a 13.7 kHz systematic modulation bandwidth.

OCTOPUS modules take on the primary computational tasks in the HyArch PIC. Figure 3a shows the top-level diagram of the OCTOPUS module, capable of performing a 5-dimensional optical dot-product operation. Within the OCTOPUS module, high-precision multiplication tasks are executed by five links (L0-L4), with passive beam splitter trees facilitating splitting and combining operations on each link. The reference link at the bottom supports coherent detection, enabling the realization of negative dot product operations and providing the bias term in the linear neuron. The output of the

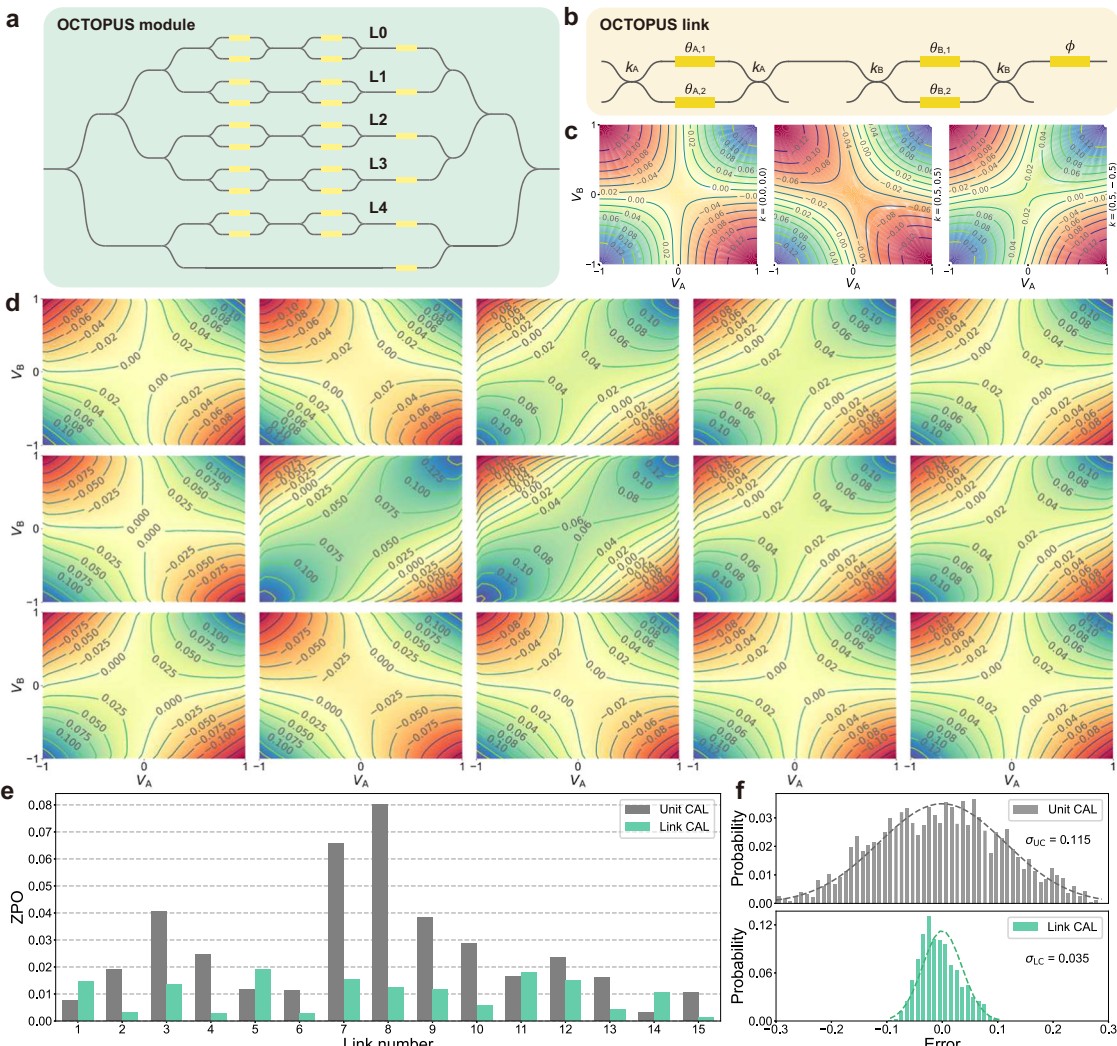

**Fig. 3 | OCTOPUS module calibration and optical dot product test. a** Concept diagram of the OCTOPUS module in the HyArch PIC. **b** Schematic of the OCTOPUS link with two push-pull MZI units and a tail phase shifter. **c** Simulated A-B joint spectra for different imbalance factors ($k_A$, $k_B$) at (0, 0), (0.5, 0.5), and (0.5, −0.5). **d** Measured A-B joint spectra $I(V_A, V_B, \lambda_{opt})$ for all OCTOPUS links under the optimal wavelength $\lambda_{opt}$ of 1530.7 nm, arranged from M0.L0 to M2.L4. **e** Distribution of ZPO for 15 OCTOPUS links after unit calibration (gray) and link calibration (green). **f** Error distribution histogram of the 10000 3-dimensional optical dot product operations calibrated by unit (gray) and by link (green).

OCTOPUS module can be expressed as:

$$I_{out} = \left\| \sum_{i}^{L} V_A^{(i)} V_B^{(i)} \widetilde{E}_i + \widetilde{E}_{ref} \right\|^2 = (\mathbf{W}\mathbf{x} + b)^2 \qquad (1)$$

where $V_A^{(i)}$ and $V_B^{(i)}$ are encoded values on the first and second MZI in the $i$ th row, and $L$ is the number of encoding links. We achieve coherent detection through intensity-based methods by leveraging the reference link and inactive encoding links, allowing the OCTOPUS module to perform a dot product operation over the entire real number domain. Equation (1) illustrates that the OCTOPUS model is equal to a general linear neuron with quadratic nonlinear activation function, and the dot product operation can be extended to general-purpose matrix-vector multiplication by encoding the matrix **W** row-wisely while keeping **x** unchanged. Figure 3b demonstrates the schematic diagram of the OCTOPUS link, comprising two push-pull MZI units and a tail phase shifter. Achieving stable multi-channel coherent inference necessitates the use of a push-pull structured MZI due to its inherent phase stability. In addition, the tail phase shifter compensates for the phase of each link in OCTOPUS module, ensuring the accuracy of the summation operation.

To enhance encoding precision in the optoelectronic computing system for maximal parallel computation, we propose link calibration —a technique involving modeling and calibrating of the entire OCTOPUS link, inspired by global nonlinear optimization[40–42] and local cell calibration[43] (Supplementary Section 2). Through link calibration, we dynamically and deterministically program the HyArch PIC in real-time without the need for optimization tailored to specific data. By conducting a two-dimensional scan of the normalized encoding values, $V_A$ and $V_B$, for push-pull MZIs A and B on a single link and measuring the corresponding normalized output light intensity, we obtain the joint spectrum $I(V_A, V_B)$. The A-B joint spectrum allows a comprehensive evaluation of link calibration effectiveness for each OCTOPUS link, providing an intuitive representation of the impact of imbalanced factors in the MZI splitter (labeled as $k$) and nonlinear mutual coupling effects. This information, challenging to discern through unit calibration alone, is quantified by evaluating the zero-point opening (ZPO = min |$I$|/(max |$I$| − min |$I$|)) as a performance metric for each link.

Figure 3c demonstrates the numerical A-B joint spectra for the three cases of balance $k_A = k_B = 0$, in-phase imbalance $k_A = k_B = 0.5$ and anti-phase imbalance $k_A = -k_B = 0.5$. In the balance model, the ZPO remains at zero. ZPO amplifies with increasing imbalance, indicating reduced unit calibration accuracy near the link zero point. Following the simulated annealing algorithm and wavelength optimization, we conducted measurements of the joint spectra $I(V_A, V_B)$ for the OCTO-PUS links based on unit calibration at the optimal wavelength of 1530.07 nm. The obtained spectra, arranged from M0.L0 to M2.L4, are depicted in Fig. 3d. Through detailed unit calibration, almost half of the links adhere closely to the balance model. However, some links still show noticeable imbalance, underscoring the importance of link calibration for accurate high-dimensional optical dot product calculations. The distribution of ZPO for 15 OCTOPUS links after unit calibration and link calibration is shown in Fig. 3e. Notably, the ZPO distribution from link calibration is significantly lower than that resulting from unit calibration, attaining nearly balance calibration for all 15 links (see Fig. S4). We implement 10000 random 3-dimensional dot-product calculations on unit-calibrated and link-calibrated OCTOPUS modules, comparing their computation errors as illustrated in Fig. 3f. The reduction in the standard deviation of normalized error across all 10,000 dot-product operations, from 0.115/2.114 = 0.0544 to 0.035/2.114 = 0.0166, substantiates the crucial role of link calibration in enabling high-dimensional and high-precision optical computation.

## Q-learning theory and cliff walking task with PIC-RL

Q-learning, a model-free, off-policy, and temporal-difference learning approach in RL, is employed to learn optimal policies by estimating the action-value function (Supplementary Section 3). The symbol "Q" in Q-learning denotes the action-value function, indicating the expected cumulative reward for a given state-action pair. This value is computed through the iterative application of the Bellman optimality operator as follows:

$$Q(s_t, a_t) \leftarrow Q(s_t, a_t) + \alpha \left( r_t + \gamma \cdot \max_a Q(s_{t+1}, a) - Q(s_t, a_t) \right) \quad (2)$$

where $r_t$ is the reward of action-state pair $(s_t, a_t)$, $\alpha$ is the learning rate and $\gamma$ is the discount factor. In the algorithm's initialization phase, the Q-table is set to zero, and during training, each cell within the Q-table is updated based on Eq. (2). Here, an "episode" refers to a single iteration of the training process, encompassing a finite number of steps. In Q-learning, the agent interacts with the environment during an episode, making decisions and updating the Q-table based on its experiences. It's important to note that achieving convergence and an optimal policy often requires multiple episodes as the agent refines its strategy over time. This iterative process ensures that the Q-table converges to values accurately representing the optimal action-value function for the given environment.

As indicated in Eq. (2), optimizing the construction of the reward function can enhance Q-learning efficiency. In this study, we introduce cosine similarity into the reward function, imparting directionality and enabling agents to perceive the distance between their current state and the target state. The modified reward function, known as the similarity reward function (SRF), outperforms the constant reward function (CRF), particularly in specific scenarios. SRF describes the reward of an agent transitioning from state $s$ to $s'$ by taking action $a$ and can be formalized as follows:

$$r_{SRF}(s, a, s') = \beta \cdot sim(\mathbf{u}(s'), \mathbf{v}) - 1 = \beta \cdot \sum_{i=1}^{n} u_i(s') v_i - 1 \quad (3)$$

where $\mathbf{u}(s')$ represents the normalized state vector of $s'$, $\mathbf{v}$ represents the normalized state vector of the target state, and $sim(\mathbf{u}(s'), \mathbf{v})$ represents the cosine similarity calculation for $n$-dimension vectors $\mathbf{u}(s')$ and $\mathbf{v}$. The parameter $\beta$ serves as the similarity coefficient, constrained within the range [0,1) to ensure effective model training. When $\beta = 0$, the SRF degenerates into the CRF, denoted by $r_{CRF} = -1$ for each step, penalizing wandering behavior.

The cliff walking task, depicted in Fig. 4a, serves as an illustrative example and a benchmark to show the process of PIC-RL and the efficiency enhancement of the SRF. The objective of the cliff walking task is to reach the goal point with the maximum cumulative reward, equivalent to searching for the shortest path in the grid world, as depicted by the green arrowed route. The grid size is configured as $4 \times 12$, with the agent starting at the lower-left cell (4, 1) and the goal cell positioned at (4, 12). If the agent moves into the cliff, the agent will incur a punishment reward of $r_P = -10$ and sends it back to the start point instantly. We experimentally calculate the similarity between all grid points and the target point using two OCTOPUS links, as illustrated in the insert of Fig. 4a. The results, shown in Fig. 4b, confirm the high computational precision of the HyArch PIC, with an error standard deviation of 0.0057. To visualize the process of Q-learning, we tabulate the Q-table in Fig. 4c. According to the definition of the Q-table, each element represents the expected cumulative reward value of the corresponding action-state pair in the cliff walking task, where the action set $A = \{up, down, right, left\}$ and state set $S$ consist of 4 and 48 elements, respectively, resulting in a $4 \times 48$ matrix for the Q-table. The computation error of the Q-table is shown in Fig. 4d, with a standard deviation of 0.0115. Given that the last row mainly represents the cliff environment, we illustrate the effective $Q$ values of the first three rows as a $4 \times 36$ matrix. We use the training curve to visually depict the evolution of agents' performance throughout their learning processes. The training curves for the cliff walking task in Fig. 4e are obtained with the similarity coefficient $\beta$ set to 0.9, based on 2000 agents. Because of the low numerical error in the on-chip optical dot product operation, the experimental SRF training curve closely aligns with the numerical counterpart. The light green region highlights the impact of acceleration: SRF RL converges 110 steps faster than CRF RL, demonstrating a 30.6% relative speedup.

To demonstrate the remarkable adaptability and resilience of the PIC-RL algorithm in the context of cliff walking, we design a more complex cliff environment on a $10 \times 10$ grid world, as depicted in Fig. 4f. The standard deviation of the similarity calculation error is 0.0045, with its corresponding error map visualized in Fig. 4g (see Fig. S7 for more details). As the grid world expands and the cliff environment becomes more complex, the convergence of the optimal step count exhibits increased variability. Therefore, we modified the convergence criterion to the optimal step count plus one (specifically, 19 steps in this environment). The training curves in Fig. 4h underscore the enhanced solution efficiency achieved by the SRF approach, surpassing the CRF approach by 12.2%. This notable performance advantage emphasizes PIC-RL's effectiveness in navigating through complex scenarios. We also study the scalability of the RL algorithm and find that the training convergence speed versus environment size $n \times n$ is about $\mathcal{O}(n^2)$ for SRF and $\mathcal{O}(n^3)$ for CRF, which indicates the scalability of SRF has an exponential advantage over CRF (Supplementary Section 4).

## Perovskite materials synthesis task with PIC-RL

Here, to further validate the efficiency and universality of PIC-RL, we highlight its application in solving a sophisticated task: the synthesis of perovskite materials. Specifically, we compare its performance with that of the original Ruddlesden-Popper (RP) phase transition metal perovskite chalcogenides $Ca_3Sn_2S_7$ (CSS)[44–46]. By partially substituting the chalcogen anion S with oxygen elements O, resulting in a general formula of $Ca_6Sn_4S_{14-x}O_x$ ($CSSO_x$, $x$ from 1 to 5), the synthesized materials exhibit enhanced performance[47]. Through theoretical analysis, we designate $CSSO_4$-0980 as the target structure for its optimal

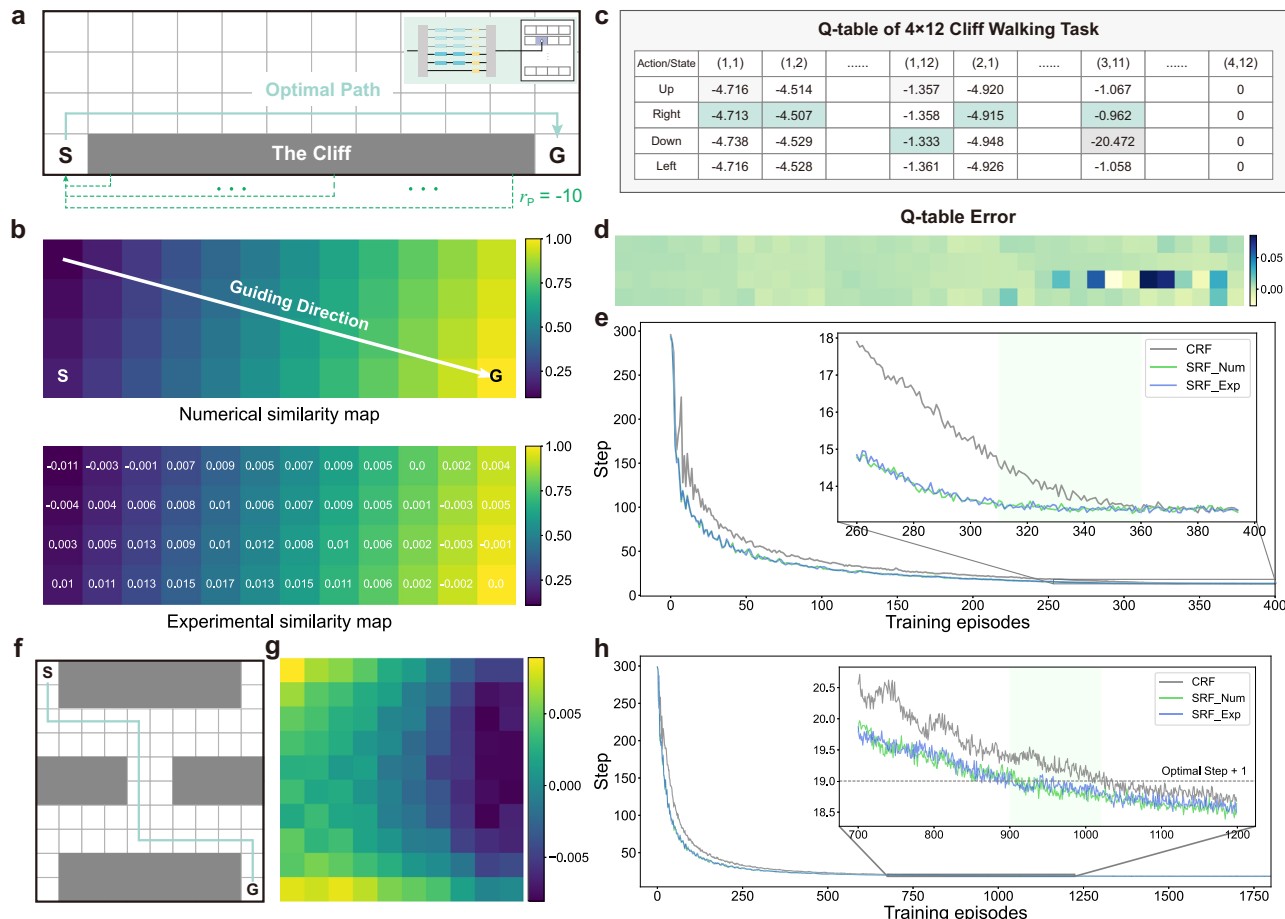

**Fig. 4 | Cliff walking task with PIC-RL. a** The 4 × 12 cliff walking grid world task includes a starting point (S) and goal point (G). The agent seeks the optimal path marked by the arrowed green line. Insert shows the HyArch PIC configuration for solving the cliff walk task. **b** Numerical (up) and experimental (down) similarity map. The white arrow indicates the guiding direction. White numbers represent calculation errors in the similarity map. **c** Q-table of the 4 × 12 cliff walking task, displaying cumulative rewards for all state-action pairs. **d** Error map between the numerical and experimental Q-tables. **e** Training curves for the 4 × 12 cliff walking task based on 2000 agents. The SRF RL algorithm improves by 30.6% over the CRF RL algorithm. **f** A 10 × 10 grid world with a complex cliff environment and its optimal path. **g** Error map between numerical and experimental similarity calculation in the 10 × 10 grid world. **h** Training curves for the RL algorithm in the complex cliff environment, with 2000 agents, indicating a 12.2% improvement for the SRF RL algorithm over the CRF RL algorithm.

performance, while the notably inefficient derivative structure CSSO₃-072 is selected as the starting structure for this RL task (Supplementary Section 5). Analogous to the high-dimensional space cliff walking task, the objective of this task is to identify the optimal synthesis route from the starting structure to the target structure, within the state space composed of all 3472 $CSSO_x$ derivative structures.

The schematic of the $CSSO_x$ crystal is shown in Fig. 5a. The left 3D crystal structure corresponds to $CSSO_1$-00, with replaceable atoms marked in different colors on the right crystal structure. We represent crystal structures using their $c$-axis coordinates as atom vectors for distinction. When replacing $x$ S atoms with O atoms, the final $x$ bits of the vector are utilized to denote the position of the O atom, while the initial 14−$x$ bits indicate the position of the S atom. This encoding method efficiently utilizes 14 non-negative numbers to differentiate between distinct structures. (All atom vectors data is provided in Supplementary Data 1). Figure 5b illustrates the optimal synthesis route, where the gray spheres represent S atoms, and the blue spheres represent O atoms. Importantly, two essential constraints govern the synthesis process: (1) Each step involves the precise replacement of a single atom, alternating between S and O. (2) Ensure that the external energy remains positive throughout the synthesis process, with the initial external energy value set to 6 eV. To compute the similarity for this task, we encode the current atom vector on the first column of

push-pull MZI modulators and the target atom vector on the second column, as depicted in the inset of Fig. 5b. With precise link calibration, the HyArch PIC enables the computation of the 14-dimensional structural similarity of perovskite materials through a single optical dot product operation. The experimental results, including all 3472 cosine similarities, are presented in Fig. 5c, along with a comparison to the numerical results. The residual error follows a normal distribution with a standard deviation of 0.015, indicating precise optical dot product calculations in high dimensions by the HyArch PIC.

Benefiting from high-fidelity R-table construction, PIC-RL agents exhibit excellent learning performance. The training curves in Fig. 5d depict the mean value (curve) and standard deviation (shaded area) derived from different reward functions, focusing on cumulative reward $V$ and convergence steps in each eposide during training. These results, obtained from 2000 agents under a similarity coefficient $\beta = 0.5$, showcase the precision of similarity calculations by the HyArch PIC. Both numerical SRF agents (blue curve) and experimental SRF agents (green curve) closely align, outperforming CRF agents (gray curve). It is remarkable that SRF RL agents achieve training convergence in 700 episodes, compared to 1600 episodes for CRF RL agents, representing a 56.25% increase in efficiency. The final Q-learning result is presented through a two-dimensional t-SNE[48] embedding of the representations in Fig. 5e. This visualization

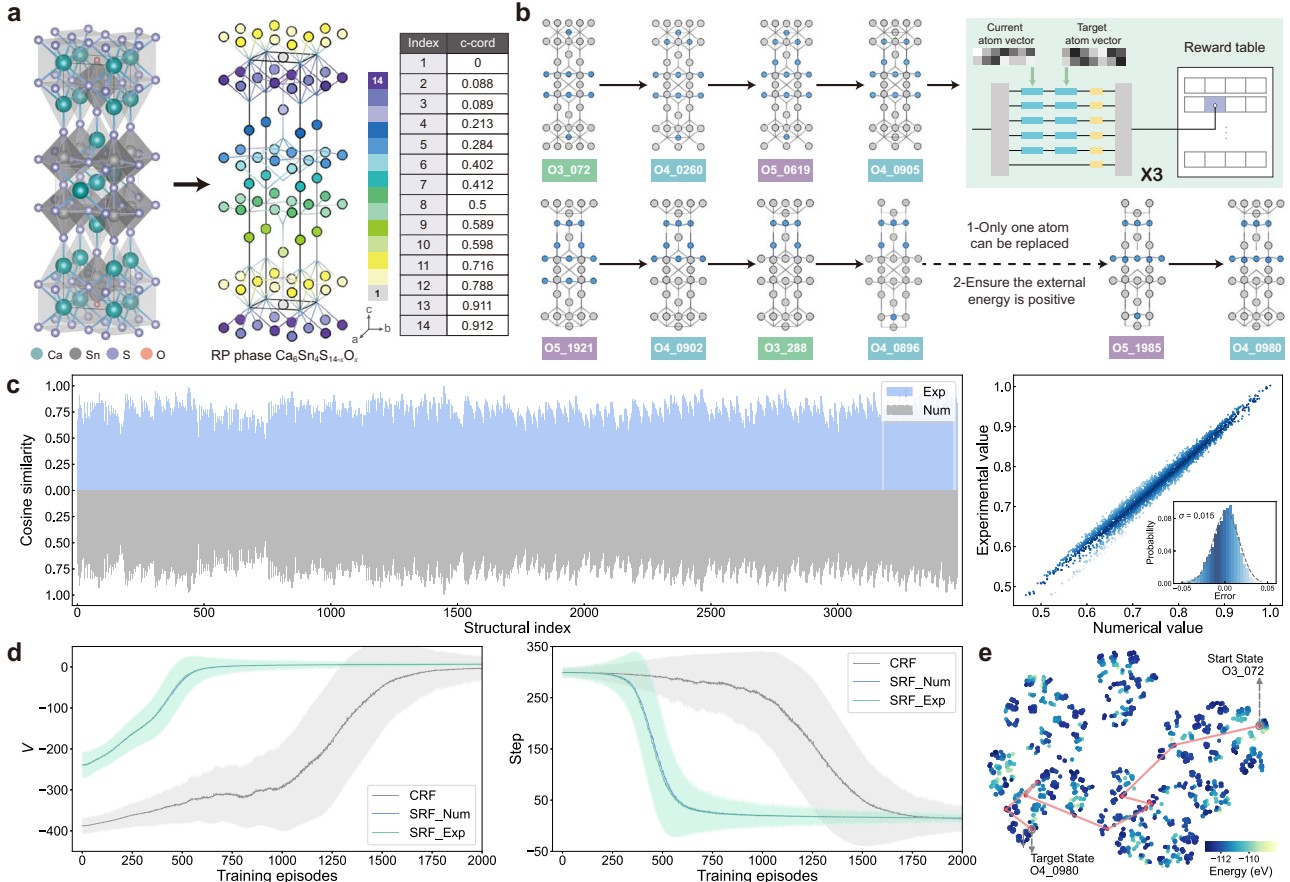

**Fig. 5 | Perovskite materials synthesis task with PIC-RL. a** Schematic of the crystal structure of perovskite with layered RP phase $Ca_6Sn_4S_{14-x}O_x$ ($x = 1$). The original crystal structure is on the left, and the positions of the replaceable atoms (S/O atoms) are indicated by circles of different colors on the right. Atom vectors are extracted from the $c$-axis coordinates in the 3D RP phase structure. **b** Optimal synthesis route in the perovskite materials synthesis task, with blue circles representing oxygen atoms and grey circles representing sulfur atoms. All 3472 derivative structures constitute the state space in this RL task. Insert shows the HyArch

PIC configuration for solving the perovskite materials synthesis task. **c** Experimental and numerical results of 3472 cosine similarity calculations. The histogram of residual errors depicts a standard deviation of 0.015. **d** Training curves for the perovskite materials synthesis task with 2000 agents, displaying cumulative reward ($V$) and convergence steps. Each curve shows the mean value, with the shaded area indicating the standard deviation. **e** t-SNE embedding of the representations to 3472 derivative structures. The red-coloured route displays the test result of the well-trained model, consistent with the synthesis route shown in **b**.

effectively projects the similarity among all 3472 structures of $CSSO_x$ onto a planar coordinate system, with the red arrowed route consistent with the optimal synthesis route depicted in Fig. 5b. The scatter dot diagram's colormap represents the total energy of each structure, and the optimal synthesis route consists of 11 intermediate states, requiring 12 structure transformations. Notably, a comparative experiment using the SRF approach to address the perovskite material synthesis task within both the Q-learning and SARSA frameworks reveals a significant 46.2% improvement in solution efficiency, confirming the superiority of Q-learning over SARSA in this SRF RL task (Supplementary Section 6).

## Discussion

HyArch PIC integrates unitary MZI mesh architecture and coherent linear neuron architecture into a monolithic PIC framework, thereby enhancing its capabilities in a noteworthy manner. This hybrid architecture offers several distinct advantages over single architectures, including: (1) high scalability and robust fault tolerance, (2) versatile functionality and (3) high-speed compatibility. We employ cosine distance $\mathcal{D}$ to assess the scalability and fault tolerance of various PIC architectures. Finite precision analysis reveals that the $N$-dimensional HyArch PIC and singular value decomposition (SVD) based mesh architectures[17] exhibit cosine distances $\mathcal{D}_{H(N)} \sim 2\sqrt{N} \log(N)\sigma_{BS}^2$ and $\mathcal{D}_{SVD} \sim 4N\sigma_{BS}^2$, respectively (Supplementary Section 7). This indicates

that HyArch PIC exhibits a sub-exponential advantage over SVD architecture PIC concerning scalability and fault tolerance. The overall transmission matrix of the $N$-dimensional HyArch PIC $\mathbf{T}_{HyArchPIC}$ (composed of an $M$-dimensional U mesh and $M$ OCTOPUS modules, where $N = M^2$) can be expressed as follows:

$$\mathbf{T}_{HyArch\ PIC} = \mathbf{T}_{U(M)}\mathbf{T}_{O(M)} = \mathbf{W}_{M\times M}\begin{bmatrix}\mathbf{u}_1 \cdot \mathbf{v}_1 \\ \vdots \\ \mathbf{u}_M \cdot \mathbf{v}_M\end{bmatrix} = \mathbf{W}_{M\times M}\begin{bmatrix}\sum_{i=1}^{M} u_{1_i}v_{1_i} \\ \vdots \\ \sum_{i=1}^{M} u_{M_i}v_{M_i}\end{bmatrix} \quad (4)$$

where $\mathbf{W}_{M\times M}$ represents the weight matrix provided by the front U module, and the $M$ OCTOPUS modules perform $M$ sets of $M$-dimensional dot product operations $\mathbf{u}_m \cdot \mathbf{v}_m$, $m = 1, 2, \ldots, M$. Equation (4) illustrates that functionally, the HyArch PIC can deploy the weighted group dot product/MVM, a critical element in advanced AI algorithms, including weighted multi-core convolution for computer vision[49], multi-head attention in natural language processing[3], and others. In $N$-dimensional optical dot product tasks, the HyArch PIC demonstrates superiority, requiring only approximately $1/N$ of the modulation units compared to the MZI mesh architecture (Supplementary Section 7). This substantial reduction simplifies integration with high-speed

electric drive, bringing optoelectronic computing chips closer to contemporary commercial GPUs in computing power. Furthermore, the HyArch PIC provides notable modularity, allowing for a more flexible layout design compared to one-way expansion MZI mesh and OCTOPUS architectures.

The optoelectronic computing chip performs MVM and dot product operations, aiming for increased computational speed and reduced energy consumption. The integration of high-speed electro-optic modulators (EOM)[50–52] and micro-electromechanical systems (MEMS)[53] further enhances the computing performance of the HyArch PIC. Optimizing the thermal optical modulator in the HyArch PIC with EOM and MEMS modulators, and adopting a configuration with $M = 128$ at a system frequency of $f_s = 10$ GHz, the number of operations per second (OPS), represented as $R \sim 2Nf_s = 2M^2f_s$, for the HyArch PIC becomes comparable to that of the state-of-the-art GPU (NVIDIA A100). Energy consumption analysis of HyArch PIC indicates that a significant portion of energy is utilized in high-speed digital-to-analog conversion (DAC). Reducing this energy consumption is crucial for the development of future optoelectronic hybrid computing architectures (Supplementary Section 8).

The introduction of cosine similarity into the reward function highlights its effectiveness in training RL models within finite discrete environmental spaces. Additionally, the technology of inverse reinforcement learning (IRL) offers an opportunity for further optimizing the reward function to enhance algorithm efficiency, representing a key research direction in the RL domain. When dealing with RL tasks featuring continuous state/action spaces, deep Q-Network (DQN)[54,55] utilizes a neural network to approximate the Q-function, effectively replacing the Q-table. Optoelectronic co-integration technologies, including on-chip digital logic and nonlinear units, are poised to significantly enhance the capabilities of photonic computing architectures[20,23,56,57]. This advancement is expected to promote the further development of PIC-RL and enable the feasibility of PIC-based DQN (Supplementary Section 9). Combining with reservoir computing provides a promising approach for substantially reducing the model parameters needed to construct DQN, offering an efficient method for integrating DQN into optoelectronic co-integration systems[58].

In conclusion, our study provides a compelling demonstration of the effectiveness of a high-dimensional PIC-assisted RL algorithm, showcasing remarkable efficiency in handling complex tasks. Leveraging our highly integrated optoelectronic computing system, we successfully achieve high-dimensional and high-precision optical dot-product computing through the integration of optimization algorithms and link calibration. The reformulation of the reward function with the similarity function greatly accelerates the convergence of Q-learning training. Subsequently, the application of PIC-RL proves instrumental in efficiently executing tasks involving cliff walking and perovskite materials synthesis. Our work establishes a generic framework for employing PIC to simulate the pivotal agent-environment interaction in RL, demonstrating a substantial acceleration effect. Furthermore, the unique advantages of the HyArch PIC open new avenues for optical neural networks and optical quantum computing. This research lays the groundwork for further exploration of RL and the implementation of more advanced AI algorithms utilizing PIC technology.

## Methods
### Fabrication and packaging of the HyArch PIC
The layout of HyArch PIC is developed and verified in LUCEDA IPKISS. Fabrication is carried out through a standard 180-nm CMOS processes on the silicon nanophotonics platform. The size of the silicon waveguide is 220 nm × 500 nm to ensure a single-mode condition. The compact footprint of HyArch PIC measures 5 mm × 5 mm. The propagation loss of HyArch PIC is <2 dB/cm and the grating coupling loss is

3.45 dB/port. To enable efficient light input and output of the HyArch PIC, we package eight grating couplers with an 8-channel standard single-mode fiber array (SMF28-FA). 174 electrical pads are wire-bonded to a printed circuit board (PCB) and linked to a homemade 256-channel electrical driver for controlling the power of 87 on-chip thermal phase shifters, with a maximum refresh rate of 100 kHz. 33 on-chip MZIs and 21 on-chip phase modulators are tuned using 200 μm long thermal phase shifters, which change the refractive index of the waveguide by local heating. Temperature stability is ensured by a dedicated temperature controller (TEC).

### Experimental setup details
The light source is a C-band tunable continuous wave (CW) laser with a maximum power of 12 dBm (Santec TSL-710C). The laser output is directed to a fibre polarization controller (FPC) and then splits into two parts: the signal light and the reference light, facilitated by a beam splitter (BS). The signal light is coupled into the HyArch PIC through FA, and the output signals from the HyArch PIC are detected by a pigtail PIN photodetector (PD) array and collected by a data acquisition module. The inclusion of reference light serves to mitigate signal jitter caused by external factors, such as mechanical vibrations and temperature fluctuations, enhancing the precision of our detection outcomes. A pigtail InGaAs PIN PD array as the receiver is used to realize integrated on-board photodetection, but its detection accuracy is limited, and the multi-channel optical power meter (Santec MPM-210) is used for high-precision photoelectric detection, such as calibration. The small form-factor pluggable (SFP) port of the development board can further integrate the transmitter of the optical module to realize end-to-end on-board optical multiply-accumulate operations. All measurements are implemented in standard room ambient conditions.

### The theory of push-pull MZI modulator
The transfer function of a single push-pull MZI modulator can be expressed as:

$$E_{out} = E_{in}\left(\cos\left(\frac{\pi}{2}\frac{\Delta_P}{P_\pi}\right) - jk\sin\left(\frac{\pi}{2}\frac{\Delta_P}{P_\pi}\right)\right)e^{j\left(\frac{\pi\Sigma_P}{2P_\pi}\right)} \qquad (5)$$

where $P_\pi$ represents the half-wave power of the MZI, $\Sigma_P = (P_A + P_B)/2$ denotes the average modulated power of the upper and lower arms, while $\Delta_P = (P_A - P_B)/2$ denotes power difference between the two arms. When $k = 0$, with the average power $\Sigma_P$ held constant, Eq. (5) reduces to the real number term multiplied by a fixed phase. This implies that an ideal push-pull MZI modulator, featuring an exact 50:50 splitter ratio, can achieve pure intensity modulation without altering the phase. Therefore, in principle, two cascaded push-pull MZIs can be employed to accomplish arbitrary multiplication of two real numbers. The tail phase shifter for each link is designed to compensate for the intra-link phase to achieve the coherent superposition between links.

### Training details of reinforcement learning models
The environment of the cliff walking task is a built-in grid world environment in the Gym framework. The dataset for perovskite materials synthesis is sourced from ref. 47. Perovskite material structures adhere to substitution structure design rules, and their energies are determined through density functional theory (DFT) calculations. Agent parameters, including the learning rate $\alpha$ and discount factor $\gamma$, are configured as 0.01 and 0.9, respectively. To improve search efficiency, we restrict the maximum number of exploration steps to 300 within each episode. For a better exploration of the environment, Q-learning uses the $\epsilon$-greedy method for agent training, allowing a

probability of $\epsilon$ to randomly select actions:

$$\pi_{act}(s) = \begin{cases} \arg\max\limits_{a \in A} Q_{opt}(s,a), & \text{prob} = 1 - \epsilon \\ \text{random from } A(s), & \text{prob} = \epsilon \end{cases} \quad (6)$$

where $\pi_{act}(s)$ is the exploration policy under state $s$. Equation (6) represents the $\epsilon$-greedy algorithm, exploring with a probability of $\epsilon$ and exploiting with a probability of $1-\epsilon$. In the early stage of model training, increased exploration is necessary to find the target. With the growing number of training episodes, the exploration rate should be continuously reduced to achieve rapid convergence of the model. Therefore, the value of $\epsilon$ decreases by $\Delta\epsilon$ in each episode. The initial $\epsilon$ value $\epsilon_0$ is set to 0.9, with an $\epsilon$ decrement of $\Delta\epsilon_{CW} = 1 \times 10^{-3}$ in cliff walking (CW) and $\Delta\epsilon_{MS} = 5 \times 10^{-6}$ in material synthesis (MS).

## Data availability
The data in the perovskite materials synthesis task comes from https://doi.org/10.1063/5.0022007, and the original structural data are available on https://github.com/j2hu/MATGANICSS. To visualize the three-dimensional structure, the perovskite materials structure file can be opened using VESTA software (https://jp-minerals.org/vesta/en/). The processed structure data for high-dimensional PIC encoding are available at Supplementary Data 1.

## Code availability
We use the Gym library for all reinforcement learning tasks. The cliff walking task is modified from the gym library case, and its source code can be found at https://www.gymlibrary.dev/environments/toy_text/cliff_walking/. The code required for implementing the perovskite materials synthesis task is available from the corresponding authors upon request.

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

## Acknowledgements

This research is supported by the National Key R&D Program of China (Grants no. 2019YFA0308703, No. 2019YFA0706302, and No. 2017YFA0303700); National Natural Science Foundation of China (NSFC) (Grants No. 62235012, No. 11904299, No. 61734005, No. 11761141014, and No. 11690033, No. 12104299, and No. 12304342); Innovation Program for Quantum Science and Technology (Grants No. 2021ZD0301500, and No. 2021ZD0300700); Science and Technology Commission of Shanghai Municipality (STCSM) (Grants No. 20JC1416300, No. 2019SHZDZX01, No.21ZR1432800, and No. 22QA1404600); Shanghai Municipal Education Commission (SMEC) (Grants No. 2017-01-07-00-02-E00049); China Postdoctoral Science Foundation (Grants No. 2020M671091, No. 2021M692094, No. 2022T150415). X.-M.J. acknowledges additional support from a Shanghai talent program and support from Zhiyuan Innovative Research Center of Shanghai Jiao Tong University.

## Author contributions

X.-M.J. supervised the project. X.-K.L. and X.-M.J. designed, simulated and laid out the photonic chip. X.-K.L., X.-Y.L. and J.-J.H implemented the algorithm and conducted the numerical experiments. X.-K.L. and J.-X.M. conducted the measurements and analyzed the data. X.-K.L., C.-Y.D, F.-K.H. and X.T. conducted the architecture performance analysis. X.-M.G. illustrated and rendered the 3D conceptual diagram. X.-K.L. and X.-M.J. wrote the paper with input from all the other authors.

## Competing interests

The authors declare no competing interests.
