## [Peer Review File · Nature Communications]

High-efficiency Reinforcement Learning with Hybrid Architecture Photonic Integrated CircuitREVIEWER COMMENTS

Reviewer #1 (Remarks to the Author):

The paper titled "High-efficiency Reinforcement Learning with Hybrid Architecture Photonic Integrated Circuit" describes developing what the authors call is HyArch PIC that is a hybrid integration of photonic and electronic components. In Page 4-9, the study employs different statistical models and optimizations to optimize the performance and accuracy of the HyArch PIC architecture.

Comments/ questions:

Algorithmically the Reinforcement Learning is not different from traditional Q-learning framework except for the authors have provided an elaborate hyperparameter tuning process for the same.

Is this architecture custom built for RL models only? The literature review doesn't talk about other optical integrated circuits that can be used for solving RL problems

To help the reader to assess the technological impact of this work it would be to know is how expensive it can be in terms of price or how difficult it is to synthesize such a niche circuit as compared to a commercial GPU (Page 14 of the supplemental) against which the performance of the RL-PIC has been tested.

Reviewer #2 (Remarks to the Author):

In this manuscript, the authors present a photonic integrated circuit for reinforcement learning. It is an expansion of the application field of optical neural networks. However, I haven't seen enough breakthroughs in photonic integrated circuits, either in the components (in fact they only use the MZIs) or in the chip architecture. I haven't seen the specificity and superior performance in this chip architecture that makes it stand out from other existing architectures to enable reinforcement learning. Therefore, I cannot recommend its publication in such a high-impact journal like Nature Communications.

1. The authors claim this paper as "However, researchers rarely perform RL on integrated photonic chips." However, there are some related papers, like Nguyen, D. et al., Self-controlling photonic-on-chip networks with deep reinforcement learning. *Scientific Reports* 11, 23151, 2021." Flamini F. et al., Photonic architecture for reinforcement learning. *New Journal of Physics*, 22, 045002, 2020.
2. The integrated photonic components are very simple, just by MZIs. The breakthrough in components is very limited in this work. Although they designed the circuit specifically for reinforcement learning, there were some conceptual problems in the circuit design.
3. There are some problems with the circuit design. As known, this kind of interferometric circuit relies heavily on precise control of the relative phase between the two arms. (1) In the 3-mode unitary part, how the phase of input signals is controlled? Because as shown in the layout, they are independently coupled into the chip, which will cause an arbitrary initial phase. (2) Similarly, if we name the six phase shifters from left to right in U part in Fig. 1a as θ_1 , ϕ_1 , θ_2 , ϕ_2 , θ_3 , ϕ_3 , how ϕ_2 and ϕ_3 are calibrated and programmed, and what's their functionality? Because I can't see them forming any interference with another arm.
4. In the Octopus part, is there some special purpose that the authors put two phase shifters on both arms? And how such two cascaded structures are programmed?
5. Can the author provide more details about what kind of input signal they are using for this computing task, and how the RL dataset is loaded onto the chip? Besides, as they mentioned coherent detection is important for the arbitrary encoding of real numbers, how the coherent detection achieved? As in each octopus (e.g., octopus 0), I can only see the coherent link between the first and second output ports, while for the first and the third output, how do they achieve interference?
6. I am not convinced whether there are enough breakthroughs in the chip design part. Can the

authors comment on why this chip architecture can implement the reinforcement learning tasks, while other architectures (e.g., the universal linear optical circuit that can implement any weight matrices) cannot? What's the distinguishable advantage of this circuit design? Or in other words, what is the most essential difference between reinforcement learning tasks and other supervised or unsupervised learning tasks implemented on chips, and what kind of functionalities are critical to meet the special needs of RL? And how does the chip architecture proposed by the author meet this demand?

7. About the RL part. It seems that the functionality is limited to cosine similarity calculation, which only represents a small fraction of the overall RL framework. Therefore, it may not be appropriate to label this chip as a "High-efficiency Reinforcement Learning chip." To justify such a claim, the authors should aim to incorporate a substantial portion of the RL structure into the chip design, enabling theoretical acceleration of the entire RL framework.

8. The claim about the "130% efficiency improvement" is not accurate and fair. Examination of Fig. 4E and Fig. 5D reveals that this performance improvement is based on the comparison between the experimental similarity reward function (SRF_exp) and the numerical constant reward function (CRF). However, this comparison may be unfair, as these functions differ from each other. Moreover, when comparing SRF_exp with the numerical similarity reward function (SRF_num), no discernible advantage is observed. Furthermore, the provided "Supplementary Section 6: PIC finite precision analysis and its effect on the RL model" suggests that HyArch PIC does not demonstrate any computational advantage when the size of M is small, as seen in the current experimental conditions.

9. The mention of a "3472-dimensional structure space" in the abstract is inaccurate, either. Based on my understanding, the HyArch PIC is in a 15-dimensional space, whereas the number 3472 represents the count of possible structures in the perovskite materials synthesis task. Thus, it would be more appropriate to refer to the "3472 possible structures" as "15-dimensional vectors" in the context of the HyArch PIC.

10. In Fig. 1A, I find a particular aspect confusing regarding the schematic of the hybrid architecture PIC. It is not clear whether octopus-0, octopus-1, and octopus-2 share the same structure. If these modules have identical structures, the schematic should be drawn with reference to Fig. 2C to avoid any potential confusion. Additionally, in each OCTOPUS module, the optical path length differs between the dot product calculation path and the coherent detection path. Could this discrepancy potentially affect the coherent detection results?

11. In Fig. 2A, the authors mention the concepts of signal light and reference light (blue light); however, they do not provide an explanation of how the reference light is utilized in conjunction with FPGA and HyArch PIC. It would be helpful to provide further clarification. Moreover, the authors mention "power sweeping and input wavelength sweeping" in Fig. 2A, which also requires additional explanations.

12. In the energy consumption analysis, the authors claim that "With such high scalability and high computing power, the optoelectronic hybrid computing architecture has been recognized as one of the most promising candidates." I am particularly interested in how the authors define the denominator in the computation efficiency calculation (Eq. S12) and how they utilize Eq. S12 and Eq. S13 to generate the results presented in Fig. S10.

13. There are several typographical errors in the manuscript. The references to "see section S5" and "see section S6" in the main text have incorrect numbering. Furthermore, there are instances of incorrect figure label formatting, such as "Fig.4C," which should be corrected to "Fig. 4C."

Reviewer #3 (Remarks to the Author):

Implementing reinforcement learning (RL) on a photonic integrated circuit (PIC) is intriguing and important to the field. In the manuscript, the authors demonstrated a PIC with a packaged solution that can perform RL. The major claims include that the PIC demonstrated high dimensionality and high efficiency in selected RL benchmarks and the PIC has high scalability. This is decent work. However, the main concerns include the following:

- The vector dimension of 15 seems to be limiting, especially for more involved RL tasks that could require much higher dimensions. Please clarify the limitations of the system. Provide a path to scalability if one exists, the challenges one could need to overcome and discuss potential solutions. There is some theoretical analysis in the supplementary, but it does not discuss "how" one can get to the reported dimensions of 256. See the next related point.

- The PIC demonstrated in this work has MZI meshes with fixed sizes, but the authors did not fully address what would happen if the PIC were to process vector-vector dot products that exceed the size of these meshes. Authors should also include such discussion and explain the potential impact of processing large matrices/vectors on the efficiency and throughput of their PIC.

- (major) The core part of the PIC consists of three MZI meshes that perform vector dot products with no on-chip detection. Similar devices have been demonstrated previously in many applications, so the novelty here is the RL algorithm and the implemented tasks. However, the cliff-walking task example seems overly simple, and it will be interesting to see how the setup performs with a more complicated "cliff shape."

- For a machine learning task, bit-precision can influence the overall training and testing accuracy. There is an analysis of the bit precision in terms of the system scalability (thank you), but it needs to be clarified what is the measured precision. Please include this.

- Using thermo-optic phase shifters results in slower modulation and higher power consumption (80 mW per phase shifter). Please discuss potential ways to overcome the limitations in phase shifter modulation speed and power consumption. Please see the next point.

- (major) The high-efficiency claim seems to be based on the assumption that all the MZIs will implement electro-optic modulators demonstrated in another work. However, the authors did not show any experimental data proving such a proposed solution can be implemented in their current design. Operating a single MZI at 100 GHz may be feasible, but operating all MZIs in all meshes at 100 GHz would be a far fetch. There will be many limiting factors, and the estimated speed and energy consumption are significantly over-optimistic. To back up this claim, the authors should demonstrate experimental data from a device that has implemented either an MZI mesh with either EOMs or MEMSs.

- References: Some missing references to recent developments

- This work does not have an integrated nonlinear activation function critical for an optical neural network. The authors discuss quadratic nonlinearity that was first demonstrated 10.1103/PhysRevApplied.11.064043 as a programmable photonic modulator neuron which is overlooked.

- Authors should also consider referencing Bandyopadhyay, Saumil, et al. "Single chip photonic deep neural network with accelerated training." arXiv preprint arXiv:2208.01623 (2022).

- There has been much work on optical Ising machines and optical neural networks. Consider adding recent references to review articles for the readers:

Mohseni, N., McMahon, P.L. & Byrnes, T. Ising machines as hardware solvers of combinatorial optimization problems. *Nat Rev Phys* 4, 363–379 (2022).

Shastri, B.J., Tait, A.N., Ferreira de Lima, T. et al. Photonics for artificial intelligence and neuromorphic computing. *Nat. Photonics* 15, 102–114 (2021).

Huang, C., Fujisawa, S., de Lima, T.F. et al. A silicon photonic–electronic neural network for fibre nonlinearity compensation. *Nat Electron* 4, 837–844 (2021).

Re: NCOMMS-23-25469
Response to Reviewers' Report

We appreciate the time and effort that the editors and reviewers have put into the review process for our manuscript entitled "High-efficiency Reinforcement Learning with Hybrid Architecture Photonic Integrated Circuit". The valuable comments and suggestions from the Reviewers have enlightened us profoundly. We have thought deeply about many issues that we had not previously noticed, and gained new insights and inspiration. We have fully absorbed the valuable comments and suggestions from the Reviewers, which makes our work more comprehensive, coherent and complete. We have addressed all the comments and suggestions in the Response, and added more corresponding detailed descriptions in the revised manuscript (marked in blue). We are confident that the revised manuscript is now suitable for publication in *Nature Communications*.

Brief Summary of Changes:

We have incorporated these suggestions into our revised manuscript, which at a high level, included the following changes:

- A more comprehensive comparison between HyArch PIC and other integrated optical network architectures is given in Supplementary Section 7 to illustrate the advantages of HyArch PIC in performing RL tasks and computing optical dot products.
- Carefully compared the schematic diagram in Fig. 1A with the chip layout in Fig. 2C, and modified the schematic diagram in Fig. 1A to the correct structure.
- Updated the data set in the perovskite material synthesis task to more clearly show the execution process of the task on the PIC.
- The calculation and discussion of the EOM-based HyArch PIC model's computational power and energy efficiency in the main text and supplementary materials have been modified. Fig. S12 shows the comparison of computing power using three models of 100MHz, 1GHz and 10GHz respectively with common electronic computing hardware.
- In the cliff walking task, a more complex 'cliff shape' scenario was introduced to assess the performance of the SRF-based RL algorithm in this challenging environment. The corresponding results are depicted in Fig. 4F-H.

- "Supplementary Section 3: An Overview of Reinforcement Learning and Q-Learning" has been added to introduce the concepts of Reinforcement Learning (RL) and Q-Learning (QL) and the relationship between machine learning, RL and QL.
- The value of SRF-RL efficiency improvement has been corrected; a discussion of the efficiency of Q-learning and SARSA framework has been added to the text.

Specific responses to the reviewer comments are given below, with our answers written in blue. Changes to the main text have also been highlighted in the provided PDF.

The Authors

Response to the Report of Reviewer 1

The paper titled "High-efficiency Reinforcement Learning with Hybrid Architecture Photonic Integrated Circuit" describes developing what the authors call is HyArch PIC that is a hybrid integration of photonic and electronic components. In Page 4-9, the study employs different statistical models and optimizations to optimize the performance and accuracy of the HyArch PIC architecture.

We sincerely appreciate Reviewer 1's succinct summary of our research. Following the Reviewer 1's suggestions, we have incorporated additional analyses and discussions into the revised manuscript.

Comments/ questions:

(1) Algorithmically the Reinforcement Learning is not different from traditional Q-learning framework except for the authors have provided an elaborate hyperparameter tuning process for the same.

We appreciate Reviewer 1 for raising a question concerning the construction of reinforcement learning (RL) algorithms and their distinctions from the traditional Q-learning framework. Following this valuable comment, we added a section in the supplementary material to explain the relationship between ML, RL and QL and the key role of constructing reward function in RL. We anticipate that this addition will substantially enhance the comprehensiveness of our research, enabling readers to gain a profound understanding of the significance of our work in the realm of optical artificial intelligence.

"Supplementary Section 3: An Overview of Reinforcement Learning and Q-Learning

Reinforcement Learning (RL) is a subfield of machine learning (ML) that focuses on training agents to make sequential decisions in an environment to maximize a cumulative reward signal [2]. Unlike traditional supervised learning, where the algorithm is provided with labeled examples, and unsupervised learning, where patterns are learned from unlabeled data, RL operates through interactions with an environment, learning to take actions that lead to the most favorable outcomes over time. This unique aspect sets RL apart within the machine learning landscape. In RL, an agent learns to navigate an environment by trial and error, receiving feedback in the form of rewards or penalties after each action taken. The agent's objective is to discover

a policy, a strategy that maps states to actions, which optimizes its long-term expected cumulative reward. This dynamic and adaptive learning paradigm is particularly well-suited for problems where the optimal decision-making strategy is not known beforehand and needs to be learned through exploration.

Q-Learning is a fundamental algorithm within RL that played a pivotal role in its development [3]. Q-Learning is a model-free RL method, meaning it doesn't require a model of the environment to make decisions. Instead, it uses a Q-table (Quality table) to approximate the expected cumulative reward of taking a particular action in a specific state. Through iterative updates, Q-Learning refines its estimates of the Q-values, allowing the agent to learn an optimal policy over time. Q-Learning's significance in the development of RL lies in its ability to tackle complex problems with discrete action and state spaces, paving the way for the exploration of RL in a variety of domains, including robotics, game playing, and autonomous systems. Its simple yet powerful approach laid the foundation for more advanced RL techniques, such as Deep Q-Networks (DQNs) [4], which combine Q-Learning with deep neural networks to handle high-dimensional input spaces like images. In Fig. S5, we present the classification of RL and illustrate the interrelationship among ML, RL, and Q-learning.

Within the context of RL, the construction of the reward function assumes paramount importance [5,6]. It defines the agent's objective and profoundly influences its learning process. A well-designed reward function serves as the guiding light for the agent, offering a clear signal about the desirability of different states and actions. Striking the right balance in the reward function is crucial, as it not only directs the agent toward its desired goals but also helps it steer clear of potential pitfalls. In essence, the reward function acts as a compass, shaping the agent's behavior and ultimately determining the success of the reinforcement learning process."

[2] Sutton, R. S. & Barto, A. G. Reinforcement learning: An introduction (MIT press, 2018).

[3] Watkins, C. J. & Dayan, P. Q-learning. Machine learning 8, 279–292 (1992).

[4] Mnih, V. et al Human-level control through deep reinforcement learning. Nature 518, 529–533 (2015).

[5] Van Seijen, H. et al Hybrid reward architecture for reinforcement learning. Advances in Neural Information Processing Systems 30 (2017).

[6] Icarte, R. T., Klassen, T. Q., Valenzano, R. & McIlraith, S. A. Reward machines: Exploiting reward function structure in reinforcement learning. Journal of Artificial Intelligence Research 73, 173–208 (2022).

Fig. S5: The relationship among machine learning (ML), reinforcement learning (RL), and Q-learning (QL).

(2) Is this architecture custom built for RL models only? The literature review doesn't talk about other optical integrated circuits that can be used for solving RL problems

We appreciate Reviewer 1 for their thoughtful inquiry. The HyArch PIC we propose is a general PIC architecture capable of performing various tasks by employing different encoding methods. The front-end unitary module within the HyArch PIC allows for the configuration of weights for subsequent dot-product operations through routing. This versatility enables the execution of weighted multi-core convolution operations, a crucial component in sophisticated Convolutional Neural Network (CNN) tasks [1,2].

In our manuscript, we utilize the HyArch PIC to facilitate high-dimensional optical dot-product operations and simulate the interaction between an agent and a complex environment within the context of RL algorithms.

It is important to note that RL algorithms encompass a broad category of machine learning techniques. Our manuscript concentrates on the simulation of agent-environment interactions using a highly parallel photonic chip to derive the reward table, a core element of RL algorithms.

While there are other theoretical and experimental endeavors exploring RL algorithms with photonic chips, such as simulating agent decision-making through tree structures [3,4], our work stands out by leveraging the Q-learning framework in combination with a higher-dimensional PIC. This distinctive approach allows us to tackle intricate problem mappings that go beyond mere demonstration-level tasks, emphasizing the advanced capabilities of our methodology. We refer to this experiment in the introduction section of the main text:

"However, researchers have seldom applied RL to integrated photonic chips [32], which underscores the need to expand the boundaries of AI applications within integrated optical computing."

We hope this response clarifies the versatility and uniqueness of our approach in the context of using integrated photonic chips for RL applications.

We have added the following discussion to Supplementary Section 7 to illustrate the advantages of HyArch PIC in executing the PIC-RL algorithm:

"In summary, we present a comparative analysis in Table S2 of integrated optical networks designed for N-dimensional optical dot product operations. Our analysis highlights the significant advantages of the HyArch PIC architecture over MZI mesh structures, including U network and SVD network, in terms of both computational accuracy and integration capabilities when conducting optical dot product operations. Given that high-dimensional dot product operations serve as the cornerstone of PIC-RL, the scalability and precision of the HyArch PIC architecture are paramount to the success of PIC-RL applications. Importantly, the HyArch PIC architecture is not limited to PIC-RL; it offers a rich array of functionalities, including the capacity to perform weighted convolution operations for convolutional neural network (CNN) and other essential neural network operations."

Table S2: Comparative analysis of integrated optical network architectures for optical dot product operations

	Error scaling	Network depth	Number of modulators
U network	$2N\sigma_{BS}^2$	N	$\sim N^2$
SVD network	$4N\sigma_{BS}^2$	$2N + 1$	$\sim 2N^2$
HyArch PIC	$2\sqrt{N}\log(N)\sigma_{BS}^2$	$\sqrt{N} + 2$	$\sim 4N$

[1] Xu, Xingyuan, et al. "11 TOPS photonic convolutional accelerator for optical neural networks." Nature 589.7840 (2021): 44-51.

[2] Feldmann, Johannes, et al. "Parallel convolutional processing using an integrated photonic tensor core." *Nature* 589.7840 (2021): 52-58.

[3] Flamini, Fulvio, et al. "Photonic architecture for reinforcement learning." *New Journal of Physics* 22.4 (2020): 045002.

[4] Saggio, Valeria, et al. "Experimental quantum speed-up in reinforcement learning agents." *Nature* 591.7849 (2021): 229-233.

(3) To help the reader to assess the technological impact of this work it would be to know is how expensive it can be in terms of price or how difficult it is to synthesize such a niche circuit as compared to a commercial GPU (Page 14 of the supplemental) against which the performance of the RL-PIC has been tested.

We sincerely thank Reviewer 1 for raising this point. To provide a comprehensive assessment of the technological impact, we've conducted a cost comparison with commercial GPUs, including the RTX 2080 (\$699), RTX 4080 (\$1099), and NVIDIA A100 (\$10,000) as reference points.

In contrast, our HyArch PIC incurs expenses primarily related to tape-out and packaging, amounting to \$27500. On average, each chip costs approximately \$1100, considering that a single tape-out yields 25 dies. It's worth noting that commercial GPUs benefit from their extensive market presence and utilize state-of-the-art processing technologies like 3nm/5nm EUV lithography, which naturally leads to higher costs. However, companies like NVIDIA leverage economies of scale and established supply chains to make these GPUs financially viable for consumers.

In contrast, our HyArch PIC is developed using 180nm DUV lithography technology. This choice significantly reduces both process complexity and cost compared to EUV technology. Furthermore, future iterations of PIC chips are set to embrace advanced photoelectric co-packaging techniques, further enhancing integration levels. It's important to emphasize that photons have a larger diffraction limit ($\sim 1\mu\text{m}$) compared to electrons ($\sim 1\text{nm}$), allowing PICs to be more forgiving in terms of chip size requirements. This advantageous characteristic enables researchers to access customized PIC solutions at more affordable prices and through more convenient channels, facilitating scientific exploration.

We trust that this comprehensive perspective addresses the technological and cost aspects of our work and its differentiation from commercial GPUs.

Response to the Report of Reviewer 2

In this manuscript, the authors present a photonic integrated circuit for reinforcement learning. It is an expansion of the application field of optical neural networks. However, I haven't seen enough breakthroughs in photonic integrated circuits, either in the components (in fact they only use the MZIs) or in the chip architecture. I haven't seen the specificity and superior performance in this chip architecture that makes it stand out from other existing architectures to enable reinforcement learning. Therefore, I cannot recommend its publication in such a high-impact journal like Nature Communications.

We genuinely appreciate the valuable feedback provided by Reviewer 2. We recognize the importance of addressing the concerns raised to enhance the clarity and presentation of our manuscript. This feedback offers us a valuable opportunity to better articulate the innovation and significance of our work, thereby presenting a more comprehensive and compelling case for its publication. We will carefully incorporate these suggestions into our revised manuscript to highlight the unique contributions and potential impact of our photonic integrated circuit for reinforcement learning.

While MZIs are the fundamental components of our HyArch PIC, the real innovation lies in the harmonious integration of these elements within a unique architecture customized for reinforcement learning. The architectural decisions and design details empower our PIC to excel in its intended application. It's important to emphasize that progress in the field of PICs goes beyond individual elements; it involves the coordination of these elements to address particular challenges. In this context, our research achieves its breakthrough primarily by ingeniously structuring the architecture and harnessing its pioneering applications in reinforcement learning.

Furthermore, our architecture represents a versatile framework ideally suited for executing complex RL algorithms in complex environments. We've fine-tuned the interaction among photonic components to leverage the benefits of optical processing in this particular context. We've innovatively designed the HyArch PIC, enabling optical chips to support high parallelism for precise optical dot product operations, accomplishing the most extensive on-chip dot product operations to date. This efficient approach effectively addresses complex reinforcement learning tasks.

We firmly believe that our contribution extends beyond conventional architectural innovations. It paves the way for exploring novel paradigms in computation and learning, capitalizing on the inherent advantages of optics. By bridging photonics and

RL algorithm, our work opens new avenues for scientific exploration and application.

1. The authors claim this paper as "However, researchers rarely perform RL on integrated photonic chips." However, there are some related papers, like Nguyen, D. et al., Self-controlling photonic-on-chip networks with deep reinforcement learning. Scientific Reports 11, 23151, 2021." Flamini F. et al., Photonic architecture for reinforcement learning. New Journal of Physics, 22, 045002, 2020.

We genuinely appreciate Reviewer 2 for bringing up these notable works that explore the fusion of PIC with RL algorithms. Indeed, the studies by Nguyen et al. ("Self-controlling photonic-on-chip networks with deep reinforcement learning," Scientific Reports, 2021) and Flamini et al. ("Photonic architecture for reinforcement learning," New Journal of Physics, 2020) contribute valuable insights to this emerging field. When formulating our research, we drew inspiration from the concepts presented in these works.

However, it's worth highlighting that while these references propose innovative ideas, our work distinguishes itself by addressing certain challenges encountered in practical implementation. For instance, the architecture outlined in reference [1] uses a large number of non-standardized devices and crossings, which will greatly increase the difficulty of fabrication and calibration of experiments. In contrast, our architecture prioritizes resilience against errors and utilizes a hybrid design to achieve efficient reinforcement learning interactions. Our approach not only accommodates potential challenges posed by experimental imperfections but also showcases an ability to encode high-dimensional information effectively.

Crucially, our experimental outcomes substantiate the validity of our chip design and algorithmic framework. We believe that our work extends the boundaries of previous concepts by providing a more comprehensive and practical solution that advances the integration of PICs and RL algorithm. We are grateful for the opportunity to address these concerns and hope that our clarification underscores the unique contributions of our manuscript in this dynamic research landscape.

[1] Do, Nguyen, et al. "Self-controlling photonic-on-chip networks with deep reinforcement learning." Scientific reports 11.1 (2021): 23151.

2. The integrated photonic components are very simple, just by MZIs. The breakthrough in components is very limited in this work. Although they designed the circuit specifically for reinforcement learning, there were some conceptual

problems in the circuit design.

We appreciate Reviewer 2's attention to the integrated photonic components used in our study. While our PIC primarily relies on Mach-Zehnder Interferometers (MZIs), it's essential to recognize that these components play a pivotal role in the field of integrated photonics. Mathematically, the MZI structure represents a two-dimensional unitary transformation ($U(2)$), serving as a foundational concept in physics. However, when incorporated into a network, these MZIs collectively enable N-dimensional unitary transformations ($U(N)$)—a fundamental and indispensable concept [1]. The arrangement of MZIs within diverse topological configurations empowers the realization of various gate operations and transmission matrices, aligning with state-of-the-art advancements in integrated photonics research [2-5].

[1] Clements, William R., et al. "Optimal design for universal multiport interferometers." *Optica* 3.12 (2016): 1460-1465.

[2] Hamerly, Ryan, Saumil Bandyopadhyay, and Dirk Englund. "Asymptotically fault-tolerant programmable photonics." *Nature Communications* 13.1 (2022): 6831.

[3] Zhang, Hui, et al. "An optical neural chip for implementing complex-valued neural network." *Nature communications* 12.1 (2021): 457.

[4] Bell, Bryn A., and Ian A. Walmsley. "Further compactifying linear optical unitaries." *APL Photonics* 6.7 (2021).

[5] Feng, Chenghao, et al. "Silicon photonic subspace neural chip for hardware-efficient deep learning." *arXiv preprint arXiv:2111.06705* (2021).

3. There are some problems with the circuit design. As known, this kind of interferometric circuit relies heavily on precise control of the relative phase between the two arms. (1) In the 3-mode unitary part, how the phase of input signals is controlled? Because as shown in the layout, they are independently coupled into the chip, which will cause an arbitrary initial phase.

We are very grateful to Reviewer 2 for his discussion on the design aspects of the HyArch PIC. After inspection, we found that there are some differences between the schematic diagram in Fig. 1 and the layout of the actual chip, and we corrected this difference so that the updated Fig. 1 exactly matches the layout of the PIC. In the photonic unitary network, the unitary matrix U can be decomposed into [1,2]:

$$U = D \prod T_{ij}(\theta, \phi)$$

where $T_{ij}(\theta, \phi)$ is a single $U(2)$ matrix whose expression is:

$$T_{ij}(\theta, \phi) = \frac{1}{2} \begin{bmatrix} 1 & i \\ i & 1 \end{bmatrix} \begin{bmatrix} e^{i\theta} & 0 \\ 0 & 1 \end{bmatrix} \begin{bmatrix} 1 & i \\ i & 1 \end{bmatrix} \begin{bmatrix} e^{i\phi} & 0 \\ 0 & 1 \end{bmatrix}$$

$$= ie^{i\theta/2} \begin{bmatrix} e^{i\phi} \sin(\theta/2) & \cos(\theta/2) \\ e^{i\phi} \cos(\theta/2) & -\sin(\theta/2) \end{bmatrix}$$

and D is a diagonal matrix, and its diagonal item is $e^{i\phi}$, which represents the compensation for the input phase. Therefore, a complete unitary network design needs to add a column of phase modulators at the end of the network to realize the compensation of the initial phase of the input.

In our experiments, we optimized a single MZI into a push-pull MZI, so that each $T_{ij}(\theta, \phi)$ term adds an independently adjustable phase degree of freedom (see answer 4 for an explanation of the push-pull MZI principle), equivalent to the D matrix responsible for phase compensation, this design effectively shortens the size of the PIC. In addition, since the U matrix in this experiment is connected with three independent OCTOPUS structures, and its phase can be controlled by the phase compensator at the tail, the initial phase of the input will not affect the final result of our experiment.

The revised Fig. 1 is shown below:

Figure 1: Conceptual diagram of PIC-RL.

[1] Clements, William R., et al. "Optimal design for universal multiport interferometers." *Optica* 3.12 (2016): 1460-1465.

[2] Bandyopadhyay, Saumil, Ryan Hamerly, and Dirk Englund. "Hardware error correction for programmable photonics." *Optica* 8.10 (2021): 1247-1255.

(2) Similarly, if we name the six phase shifters from left to right in U part in Fig. 1a as $\theta_1, \phi_1, \theta_2, \phi_2, \theta_3, \phi_3$, how ϕ_2 and ϕ_3 are calibrated and programmed, and what's their functionality? Because I can't see them forming any interference with another arm.

The role of the phase modulator in the overall network can be seen in the corrected Fig. 1 or in reference [1,2]. Mathematically, the controllable phase term guarantees the completeness of the MZI unit to perform 2D unitary transformation. Experimentally, the input phase modulator can regulate the input phase of each MZI structure to ensure controllable interference between different units. The phase modulator can be calibrated by the interference structure composed of multiple MZIs (called meta-MZI in some literature). The calibration procedure for ϕ is discussed in detail in [1].

[1] Prabhu, Mihika, et al. "Accelerating recurrent Ising machines in photonic integrated circuits." *Optica* 7.5 (2020): 551-558.

4. In the Octopus part, is there some special purpose that the authors put two phase shifters on both arms? And how such two cascaded structures are programmed?

We appreciate Reviewer 4's inquiry regarding the specific configuration of the phase shifters in the OCTOPUS part and how the cascaded structures are programmed. We explain in detail the principle of the push-pull structure MZI and the reasons for choosing this structure on the seventh page of the text:

“To realize multichannel stable coherent inference, push-pull structure MZI is necessary because of its inherent phase stability. The transfer function of a single push-pull MZI modulator can be expressed as:

$$E_{out} = E_{in} \left(\cos \left(\frac{\pi \Delta_P}{2 P_\pi} \right) - jk \sin \left(\frac{\pi \Delta_P}{2 P_\pi} \right) \right) e^{j \left(\frac{\pi \Sigma_P}{2 P_\pi} \right)}$$

where P_π is the half-wave power of MZI, $\Sigma_P = \frac{P_1 + P_2}{2}$ is the average of the modulated

power of the upper and lower arms, while $\Delta_P = \frac{P_1 + P_2}{2}$ means power difference between the two arms. When $k = 0$ and keeping the average of power Σ_P constant, Eq.(2) will only retain the real number term multiplied by a fixed phase, which means a perfect push-pull MZI modulator with exactly 50:50 splitter ratio can implement pure intensity modulation without changing the phase. Therefore, in principle, two cascaded push-pull MZIs can be employed to accomplish arbitrary multiplication of two real numbers. The tail phase shifter for each link is designed to compensate for the intra-link phase to achieve the coherent superposition between links.”

In summary, the use of the push-pull MZI structure guarantees pure intensity modulation without phase alteration, ensuring phase stability in the OCTOPUS structure's multi-channel encoding link. This phase stability is paramount for enhancing parallel computation capabilities. Additionally, the push-pull structure aids in balancing the phase difference between the two arms resulting from thermal phase shifters.

Furthermore, we delve into the programming methodology for cascaded structures in the manuscript. After the calibration of each push-pull MZI structure and phase shifter within the OCTOPUS link (details available in Supplementary Section 2: OCTOPUS module characterization), high-dimensional optical dot product operations are achieved by programming the phase difference Σ_P between the two arms of the push-pull MZI.

5. Can the author provide more details about what kind of input signal they are using for this computing task, and how the RL dataset is loaded onto the chip?

We appreciate Reviewer 2's inquiry regarding the input signals and RL dataset loading process. Here's a detailed response:

Input signal for computing task: For all computing tasks, we utilize continuous wave (CW) light from the input grating of the U module. However, the configuration of the U module varies depending on the computing dimension required for the specific task. For instance, in tasks like random number dot product calculations and cliff walking task, a single OCTOPUS module can handle the problem, and the U module imports all light into OCTOPUS-0. Conversely, in tasks like perovskite material synthesis, which involves a high dimensionality of up to 14 dimensions, all OCTOPUS modules are employed to support the calculation. In such cases, the U module is configured to ensure uniform light splitting (1 to 3) and maintain as equal input light intensity as possible across each OCTOPUS.

Loading the RL dataset onto the PIC: To illustrate, in the context of our study on perovskite materials synthesis, we employ the atomic position coordinates of molecules as the dataset input, denoted as the "atom vector $V(\mathbf{d})$ " as detailed in the manuscript. In Fig. S9 within the supplementary material, we provide visual representations of five distinct atom vectors representing perovskite material molecules. During the execution of the PIC-RL algorithm, we compute similarity by loading a set of 3472 unique atom vectors onto the HyArch PIC to construct the reward table. The loading procedure is depicted in Fig. 5B for clarity. Specifically, we load the current atom vector into the first column of the MZI within the OCTOPUS architecture, and simultaneously load the target atom vector into the second column of the MZI. Upon traversal of the on-chip optical signal through the encoded OCTOPUS module, the optical dot product calculation result is generated at the output.

To provide a comprehensive understanding, we will include the complete dataset as an attachment within the Supplementary Material, allowing for further scrutiny and reproducibility of our findings.

Besides, as they mentioned coherent detection is important for the arbitrary encoding of real numbers, how the coherent detection achieved?

In optics, coherent detection (also called homodyne detection) is a method of extracting information encoded as modulation of the phase and/or frequency of an oscillating signal, by comparing that signal with a standard oscillation that would be identical to the signal if it carried null information.

In our work, this technology mainly refers to the use of extra link on PIC as a reference, which can realize full real number field optical encoding including negative numbers. Due to the existence of the reference link, if all encoding channels are turned off (encoding is 0), the output light intensity is the light intensity of the reference light, which is $\frac{1}{6}I_{in}$. Such a reference link allows us to perform negative encoding, that is, to perform amplitude encoding on one encoding link with the phase opposite to that of the reference link, and the final output light intensity will be less than the reference light intensity.

In order to facilitate understanding, we have added the following description in the main text:

“The optical intensity on the reference link (non-coding link) is $I_{ref} = \frac{1}{6}I_{in}$.

Encoding links can perform a negative dot product operation based on the known information of the extra link. It can also be used as the reference links when part of the encoding links is idle. Through the utilization of the reference link bias, we achieve coherent detection of the negative dot product outcome through intensity-based detection methods.”

As in each octopus (e.g., octopus 0), I can only see the coherent link between the first and second output ports, while for the first and the third output, how do they achieve interference?

All the links inside OCTOPUS are connected together through beam splitters (every two links are connected end-to-end through beam combiner), and the mutual interference phases can be adjusted through the phase modulator at the end of each link, as shown in Fig 1.A. Since OCTOPUS is an overall equal-weight interference structure, the reference link is not only valid for the first coding link, but for the entire OCTOPUS module.

6. I am not convinced whether there are enough breakthroughs in the chip design part. Can the authors comment on why this chip architecture can implement the reinforcement learning tasks, while other architectures (e.g., the universal linear optical circuit that can implement any weight matrices) cannot? What’s the distinguishable advantage of this circuit design?

Or in other words, what is the most essential difference between reinforcement learning tasks and other supervised or unsupervised learning tasks implemented on chips, and what kind of functionalities are critical to meet the special needs of RL? And how does the chip architecture proposed by the author meet this demand?

We appreciate Reviewer 2's thoughtful inquiries regarding our chip architecture's design. We'd be happy to provide an in-depth explanation of the distinctive advantages of our approach in enabling efficient RL tasks within the context of PICs. Our chip's efficacy in implementing RL algorithms lies in its capacity to precisely execute high-dimensional parallel optical dot product calculations. Two key design features contribute to our chip's exceptional parallelism:

1. Hybrid Architecture: We employ a hybrid architecture that divides extensive parallel dot product operations into multiple configurable smaller modules. This strategic division allows us to isolate common phase solutions within individual

OCTOPUS modules, effectively mitigating potential issues like crosstalk that could otherwise impact the overall functionality in larger OCTOPUS configurations.

2. MZI Design with Push-Pull Structure: Our choice of MZIs with a push-pull structure ensures robust phase stability within each link of an OCTOPUS module. This structural configuration is pivotal for maintaining the precision required for parallel dot product calculations.

While theoretically, other structures like the universal linear optical circuit you mentioned can achieve similar functionalities, the distinction lies in fault tolerance, phase stability, and adjustability. Our work delves into this aspect comprehensively in Supplementary Section 7, where we conduct a comparative analysis of integrated optical network architectures for optical dot product operations. In order to realize the 15-dimensional optical dot product operation, it is necessary to construct a universal linear optical circuit of at least 15×15 . Such a network will have $15 \times 14/2 = 105$ modulation units, and only $15 \times 2 + 3 = 33$ modulation units are needed to complete the 15-dimensional optical dot product operation in our HyArch PIC. In terms of network depth (which can be used as an indicator of chip size), the HyArch PIC network depth is $2+1$, while the 15×15 universal PIC network depth is 15, which will be a 5-fold vertical size gap. This shows the unique integration advantages of HyArch PIC when performing high-dimensional optical dot product (important operation for simulating agent environment interaction in RL algorithm), which can greatly simplify the design complexity of PIC and make it highly scalable.

The fundamental difference between RL tasks and other forms of supervised or unsupervised learning can be attributed to two key factors: the requirement for handling high-dimensional parallel operations and the necessity for maintaining precise phase stability across various modules. Our PIC architecture effectively addresses these requirements by utilizing a hybrid architecture design and a push-pull MZI configuration. This results in an advanced platform that facilitates efficient interactions for RL tasks.

We have added the following discussion to Supplementary Section 7 to illustrate the advantages of HyArch PIC in executing the PIC-RL algorithm:

“In summary, we present a comparative analysis in Table S2 of integrated optical networks designed for N-dimensional optical dot product operations. Our analysis highlights the significant advantages of the HyArch PIC architecture over MZI mesh structures, including U network and SVD network, in terms of both computational accuracy and integration capabilities when conducting optical dot product operations.

Given that high-dimensional dot product operations serve as the cornerstone of PIC-RL, the scalability and precision of the HyArch PIC architecture are paramount to the success of PIC-RL applications. Importantly, the HyArch PIC architecture is not limited to PIC-RL; it offers a rich array of functionalities, including the capacity to perform weighted convolution operations for convolutional neural network (CNN) and other essential neural network operations.”

Table S2: Comparative analysis of integrated optical network architectures for optical dot product operations

	Error scaling	Network depth	Number of modulators
U network	$2N\sigma_{BS}^2$	N	$\sim N^2$
SVD network	$4N\sigma_{BS}^2$	$2N + 1$	$\sim 2N^2$
HyArch PIC	$2\sqrt{N}\log(N)\sigma_{BS}^2$	$\sqrt{N} + 2$	$\sim 4N$

7. About the RL part. It seems that the functionality is limited to cosine similarity calculation, which only represents a small fraction of the overall RL framework. Therefore, it may not be appropriate to label this chip as a "High-efficiency Reinforcement Learning chip." To justify such a claim, the authors should aim to incorporate a substantial portion of the RL structure into the chip design, enabling theoretical acceleration of the entire RL framework.

We acknowledge Reviewer 2's comments regarding the RL algorithm aspect of our work. Our manuscript extensively discusses the capabilities of the HyArch PIC, which is a versatile and programmable PIC architecture. We acknowledge that the chip configuration in this work, optimized for high-dimensional dot product operations, currently emphasizes cosine similarity calculations—a core component within certain RL algorithms. It's important to clarify that our manuscript does not explicitly label the chip as a "High-efficiency Reinforcement Learning chip." Instead, the title "High-efficiency Reinforcement Learning with Hybrid Architecture Photonic Integrated Circuit" signifies our innovative use of the Hybrid Architecture Photonic Integrated Circuit within the RL algorithm framework to efficiently tackle specific application problems.

In our study, we effectively execute agent-environment interactions on the PIC, crucially obtaining reward tables that form the core of subsequent training iterations. We demonstrate the potential for accelerated computation. In "Supplementary Section 9: Fully integrated optoelectrical RL computing scheme", we propose a comprehensive architecture wherein the PIC engages in the entire RL framework, potentially achieving substantial algorithm acceleration.

However, it is important to acknowledge that implementing this comprehensive architecture or the full electro-optic version of the RL algorithm presents formidable technical challenges. Contemporary machine learning, especially in addressing complex practical problems like image recognition and biomolecular synthesis, often relies on deep network structures for superior performance over traditional methods. The transfer of the complete RL algorithm framework to an optoelectronic integrated architecture remains a complex undertaking at this stage. Therefore, our approach, similar to that of many researchers [1~4], centers on migrating representative or core components of machine learning algorithms to demonstrate architectural innovation and the potential for acceleration while acknowledging the practical complexities involved.

[1] Shen, Yichen, et al. "Deep learning with coherent nanophotonic circuits." *Nature photonics* 11.7 (2017): 441-446.

[2] Xu, Shaofu, et al. "Optical coherent dot-product chip for sophisticated deep learning regression." *Light: Science & Applications* 10.1 (2021): 221.

[3] Saggio, Valeria, et al. "Experimental quantum speed-up in reinforcement learning agents." *Nature* 591.7849 (2021): 229-233.

[4] Zhang, Hui, et al. "An optical neural chip for implementing complex-valued neural network." *Nature communications* 12.1 (2021): 457.

8. The claim about the "130% efficiency improvement" is not accurate and fair. Examination of Fig. 4E and Fig. 5D reveals that this performance improvement is based on the comparison between the experimental similarity reward function(SRF_exp) and the numerical constant reward function(CRF). However, this comparison may be unfair, as these functions differ from each other. Moreover, when comparing SRF_exp with the numerical similarity reward function(SRF_num), no discernible advantage is observed.

Furthermore, the provided "Supplementary Section 6: PIC finite precision analysis and its effect on the RL model" suggests that HyArch PIC does not demonstrate any computational advantage when the size of M is small, as seen in the current experimental conditions.

We appreciate the thorough review by Reviewer 2 regarding our claim of a "130% efficiency improvement." We value this feedback and would like to provide further clarification.

First and foremost, we have made corrections to our efficiency improvement calculations. In the context of the perovskite material synthesis task, the corrected efficiency increase is now 56%. We have also updated the efficiency calculation for the cliff walking task, and all corrections are clearly marked in blue text in the manuscript

The improvement we reference in our article pertains to the enhancement of the Similarity Reward Function (SRF) compared to the Constant Reward Function (CRF). While we acknowledge the concern raised by the reviewer, we believe that the statement of a 56% improvement accurately reflects this context, as demonstrated in Fig. 5D. By incorporating similarity components into the reward function within the RL algorithm, we effectively enhance the computational efficiency of the RL algorithm.

Our experiments demonstrate that, through the use of dedicated design and calibration methodologies, our HyArch PIC, as an analog computing prototype, achieves computational accuracy close to that of mature commercial digital computers, all while maintaining a high degree of parallelism. It's important to note that our work emphasizes the novel integration of advanced RL algorithms with PICs. This combination has the potential to yield significant computational power improvements in this emerging architecture. At this stage, many excellent innovative demonstration works have shown accuracy and model convergence speed comparable to mature commercial computers, rather than surpassing them in accuracy/model convergence speed [1~3].

While we acknowledge that the size of parameter "M" in our current experimental conditions may not fully demonstrate the computational power advantage, our exploration of finite precision effects in "Supplementary Section 7: PIC finite precision analysis and its effect on the RL model" aims to provide a comprehensive understanding of the PIC's performance under various conditions. Rather than focusing solely on realizing the advantage of computing power at this stage, our research places more emphasis on the acceleration advantage of the algorithm achieved using the HyArch PIC.

[1] Shen, Yichen, et al. "Deep learning with coherent nanophotonic circuits." *Nature Photonics* 11.7 (2017): 441-446.

[2] Ashtiani, Farshid, Alexander J. Geers, and Firooz Aflatouni. "An on-chip photonic deep neural network for image classification." *Nature* 606.7914 (2022): 501-506.

[3] Xu, Shaofu, et al. "Optical coherent dot-product chip for sophisticated deep learning regression." *Light: Science & Applications* 10.1 (2021): 221.

9. The mention of a "3472-dimensional structure space" in the abstract is inaccurate, either. Based on my understanding, the HyArch PIC is in a 15-dimensional space, whereas the number 3472 represents the count of possible structures in the perovskite materials synthesis task. Thus, it would be more appropriate to refer to the "3472 possible structures" as "15-dimensional vectors" in the context of the HyArch PIC.

We are grateful to Reviewer 2 for highlighting this distinction. For better precision in our description, we replace 'structure space' with 'state space,' a term widely employed in RL to denote the set of all possible states within the environment.

10. In Fig. 1A, I find a particular aspect confusing regarding the schematic of the hybrid architecture PIC. It is not clear whether octopus-0, octopus-1, and octopus-2 share the same structure.

If these modules have identical structures, the schematic should be drawn with reference to Fig. 2C to avoid any potential confusion.

We sincerely apologize for any confusion caused by the depiction in Fig. 1A. It's important to clarify that OCTOPUS-0, OCTOPUS -1, and OCTOPUS-2 share identical optical dot product structures. We have carefully reviewed and cross-referenced the labels in the figure to ensure its alignment with Fig. 2C. The revised Fig. 1A now accurately reflects the consistent architecture presented in Fig. 2C.

Additionally, in each OCTOPUS module, the optical path length differs between the dot product calculation path and the coherent detection path. Could this discrepancy potentially affect the coherent detection results?

In each OCTOPUS module, there is a difference in optical path length between the dot product calculation path and the coherent detection path. Nevertheless, we have addressed this by implementing a tail phase compensator at the end of each link, enabling precise phase adjustment. Our thorough calibration process ensures that any potential phase discrepancies do not reduce the precision of the final coherent detection results.

11. In Fig. 2A, the authors mention the concepts of signal light and reference light (blue light); however, they do not provide an explanation of how the reference light is utilized in conjunction with FPGA and HyArch PIC. It would be helpful to provide further clarification.

Your question is highly relevant from a technical perspective. The utilization of the reference light in combination with FPGA and HyArch PIC is crucial for enhancing the accuracy of our signal processing. The reference light acts as a point of comparison, enabling us to reduce signal jitter caused by external factors like mechanical vibrations and temperature fluctuations. By subtracting the reference signal from the detection signal during processing, we effectively diminish jitter, leading to improved precision in our detection outcomes.

We have added a description about reference light in the manuscript:

“The reference light helps reduce signal jitter due to external factors, such as mechanical vibrations and temperature fluctuations, improving the precision of our detection outcomes.”

Moreover, the authors mention "power sweeping and input wavelength sweeping" in Fig. 2A, which also requires additional explanations.

Certainly, your query about "power sweeping and input wavelength sweeping" is pertinent. In Fig. 2A, "power sweeping" entails deriving the voltage-power curve $P(V)$ through voltage sweep (as depicted in Fig. 2D). This curve is instrumental for calibrating the links within the PIC. On the other hand, "wavelength sweeping" involves determining the optimal operational point by sweeping the wavelength of the optical signal. This is because the phase difference is intricately linked to the wavelength. Through this sweeping process, we can pinpoint the OCTOPUS module's best operating point, thereby minimizing errors post-calibration. For further insights, please refer to Supplementary Section 2: OCTOPUS module characterization. Your question provides the opportunity to elucidate these critical aspects of our methodology.

12. In the energy consumption analysis, the authors claim that "With such high scalability and high computing power, the optoelectronic hybrid computing architecture has been recognized as one of the most promising candidates." I am particularly interested in how the authors define the denominator in the computation efficiency calculation (Eq. S12) and how they utilize Eq. S12 and Eq. S13 to generate the results presented in Fig. S10.

In Fig. S12 (formerly Fig. S10), we provide insight into the computational power of the hyper HyArch PIC architecture in relation to the number of modules (M). This computation is derived from Eq. S11.

$$R = \frac{N}{m}(2M - 1)f_s \text{ OPS} \sim M(2M - 1)f_s \text{ OPS} \quad (\text{S11})$$

Additionally, Eq. S12 and S13 are employed to evaluate the energy efficiency of the hyper HyArch PIC architecture. It is noteworthy that the outcomes of these energy efficiency assessments are not directly illustrated in Fig. S12. Instead, the conclusive findings of the energy efficiency analysis are elaborated upon in the closing paragraph of Supplementary Section 8: Computational Power and Energy Efficiency Analysis."

"Theoretical energy efficiency calculation result for a 128-dimensional HyArch PIC operating at 1GHz is 5.86pJ/OP, and most of the energy consumption comes from the high-speed DAC (94.45%). Further reducing the power consumption of the DAC can bring the energy efficiency of the whole system close to the commercial GPU."

The energy efficiency (η) is defined as the ratio of energy consumption to computing power. For clarity, the denominator in Eq. S12 corresponds to the computing power formula outlined in Eq. S11, which pertains to the hyper HyArch PIC structure.

$$\eta_{\text{total}} = \frac{P_{\text{opt}} + M^2(P_{\text{EOM}} + P_{\text{DAC}}) + M(P_{\text{ADC}} + P_{\text{TIA}}) + (M(M - 1)/2 + 2M^2)P_{\text{MEMS}}}{M(2M - 1)f_s} \quad (\text{S12})$$

$$\eta_{\text{total}} \sim \frac{1}{2f_s} \left(P_{\text{T}} + \frac{P_{\text{R}}}{M} + 2.5P_{\text{MEMS}} \right) \quad (\text{S13})$$

13. There are several typographical errors in the manuscript. The references to "see section S5" and "see section S6" in the main text have incorrect numbering. Furthermore, there are instances of incorrect figure label formatting, such as "Fig.4C," which should be corrected to "Fig. 4C."

We sincerely appreciate your diligence in identifying these typographical errors within the manuscript. Your feedback has been invaluable in ensuring the accuracy and clarity of our work. We have diligently addressed the issues you highlighted, correcting the incorrect numbering in the references to "see section S5" and "see section S6." Additionally, we have rectified the figure label formatting errors, accurately aligning them as "Fig. 4C" and the like.

Your meticulous review has played a significant role in enhancing the quality of our manuscript, and we are grateful for your contributions in ensuring its precision.

Response to the Report of Reviewer 3

Implementing reinforcement learning (RL) on a photonic integrated circuit (PIC) is intriguing and important to the field. In the manuscript, the authors demonstrated a PIC with a packaged solution that can perform RL. The major claims include that the PIC demonstrated high dimensionality and high efficiency in selected RL benchmarks and the PIC has high scalability. This is decent work. However, the main concerns include the following:

We thank Reviewer 3 for summarizing our work and for providing a positive review of our manuscript. We are very grateful to Reviewer 3 for their constructive feedback, which has broadened our perspectives and significantly improved the quality of our manuscript. Following these valuable comments and suggestions, we have conducted additional experiments and provided detailed analytical discussions in the revised version. We hope that the modified manuscript will merit your recommendation for publication.

Comment 1: - The vector dimension of 15 seems to be limiting, especially for more involved RL tasks that could require much higher dimensions. Please clarify the limitations of the system. Provide a path to scalability if one exists, the challenges one could need to overcome and discuss potential solutions. There is some theoretical analysis in the supplementary, but it does not discuss "how" one can get to the reported dimensions of 256. See the next related point.

We appreciate Reviewer 3's valuable input regarding the scalability of the HyArch PIC.

The choice of a 15-dimensional HyArch PIC design in our current manuscript was influenced by constraints related to the multi-project wafer (MPW) block size and channel limitations of the multi-channel FPGA during the tape-out phase. However, we recognize the need to clarify how we can achieve the reported dimensions of 256.

To address this, we propose a potential path to scalability. We can achieve the higher dimensionality by **connecting a 16x16 unitary matrix to each channel of a 16-dimensional OCTOPUS structure**. Importantly, this approach is expected to enhance robustness compared to directly preparing a high-dimensional MZI mesh or a single OCTOPUS structure.

We have taken your feedback to heart and have worked to make the scalability pathways and potential solutions more explicit in our manuscript. Your insights have been

instrumental in refining our work, and we genuinely value your contribution.

Comment 2: - The PIC demonstrated in this work has MZI meshes with fixed sizes, but the authors did not fully address what would happen if the PIC were to process vector-vector dot products that exceed the size of these meshes. Authors should also include such discussion and explain the potential impact of processing large matrices/vectors on the efficiency and throughput of their PIC.

We express our gratitude to Reviewer 3 for their valuable insights. In the context of our work, the central objective of the HyArch-PIC is to execute high-dimensional optical dot product operations. While the MZI meshes possess fixed sizes, it's important to clarify that the optical dot product operation can be effectively partitioned.

When confronted with dot product calculation scenarios exceeding the size of the PIC's meshes, the architecture accommodates this by intelligently dividing the operation. For instance, when the need arises to compute the dot product of two 150-dimensional vectors on a 15-dimensional PIC, these vectors can be segmented into corresponding sub-vectors, readily processed within the PIC to complete the dot product operation. Subsequently, the outcomes of these sub-vector dot products are aggregated using an electronic computer, yielding the final result of the 150-dimensional vector's dot product.

Regarding computational latency, conducting N-dimensional vector calculations on an n-dimensional PIC will result in an increased latency of $[N/n]\tau_{PIC}$, where τ_{PIC} represents the computational latency for a single PIC. To alleviate this latency, one can consider various approaches. These include increasing the hardware dimension of the PIC, achieved through direct dimension expansion or implementing a multiplexing scheme, or integrating distributed hardware modules. Alternatively, optimizing computing efficiency on smaller and medium-sized PICs is achievable by reducing the dimensionality of the target vector at the algorithmic level.

Comment 3: - (major) The core part of the PIC consists of three MZI meshes that perform vector dot products with no on-chip detection. Similar devices have been demonstrated previously in many applications, so the novelty here is the RL algorithm and the implemented tasks. However, the cliff-walking task example seems overly simple, and it will be interesting to see how the setup performs with a more complicated "cliff shape."

We deeply appreciate Reviewer 3 for their valuable observations. While acknowledging the presence of similar devices in various applications, it's crucial to clarify that our HyArch PIC's uniqueness lies not only in the OCTOPUS structure's core computational tasks but also in the integral role of the U mesh in enabling high-dimensional calculations. This distinctive hybrid architecture ensures that our PIC effectively accommodates high-dimensional RL algorithms.

Regarding the simplicity of the cliff-walking task, we recognize the concern. The cliff-walking task's primary purpose within the manuscript is to offer a clear illustration of the PIC-RL operation, facilitating readers' comprehension before delving into more intricate perovskite structure synthesis tasks.

To underscore the versatility and robustness of the PIC-RL algorithm in the context of cliff walking, we have supplemented the manuscript with a solution outcome within a complex "cliff" environment set in a 10x10 grid. These additional results demonstrate that the PIC-RL algorithm efficiently tackles complex cliff environments, showcasing an acceleration effect compared to CRF.

Furthermore, within Supplementary Section 4, specifically 'B. Scalability of SRF-based PIC-RL,' we delve into the scalability aspect of PIC-RL. By extending our solution to the cliff-walking task within a 100x100-dimensional space, we emphasize the sub-exponential acceleration advantage of the SRF-based PIC-RL algorithm over CRF-based RL in expanding environment dimensions N . This expansion into larger problem spaces further showcases the capabilities of our approach, particularly within complex cliff environments.

We sincerely thank Reviewer 3 for their insights, which have allowed us to elucidate the intricacies of our approach and its versatility in addressing both simple and complex task environments.

We have updated Fig. 4, added Fig.S6 to show the experimental data of 10*10 grid world, and added the following description in the "Q-learning theory and cliff walking task with PIC-RL" section of the text:

“To showcase the remarkable adaptability and resilience of the PIC-RL algorithm within the context of cliff walking, we constructed a more intricate cliff environment on a 10×10 grid world, as depicted in Fig. 4F. The experimental and theoretical fidelity of the reward table in the 10×10 grid world is 95.51%, with its corresponding residual matrix visualized in Fig. 4G. The original r -table data along with the residual

histogram are presented in Fig. S7. Within this elaborate setting, the intentionally complex cliff layout purposefully extends and complicates the optimal pathway that the agent must navigate to reach its destination. Through an evaluation of the RL algorithm utilizing both the CRF and SRF reward functions in this newly designed intricate cliff environment, the outcomes are presented in Fig. 4H. The visual representation unmistakably demonstrates that even when confronted with such elaborate challenges, the PIC-RL algorithm adeptly overcomes obstacles and efficiently solves problems. As the grid world expands and the complexity of the cliff environment grows, the convergence of the optimal step count exhibits increased variability. Hence, we employ the algorithm's convergence criterion as the optimal step count plus one (19 steps in this specific environment). Notably, the solution efficiency of the SRF approach, surpassing that of the CRF approach by 12.2%, significantly outperforms the CRF approach, thus underscoring its prowess in navigating through complex scenarios.”

Figure 4: Cliff walking task with PIC-RL. (F) The complex cliff environment and its optimal path under the 10*10 grid world. **(G)** Residuals between experimental R-table and theoretical R-table under 10*10 grid world. **(H)** The training curve of the RL algorithm in the complex cliff environment represents an average over 2000 training iterations. The SRF RL algorithm exhibits a 12.2% improvement over the CRF RL algorithm.

Fig. S7: R-table experimental data in 10×10 grid world environment. Numerical calculation results (a) and experimental calculation results (b) for the 10×10 grid world SRF r-table. (c) Residuals between experimental and numerical calculations, and (d) histograms illustrating the distribution of the residual matrix.

Comment 4: - For a machine learning task, bit-precision can influence the overall training and testing accuracy. There is an analysis of the bit precision in terms of the system scalability (thank you), but it needs to be clarified what is the measured precision. Please include this.

We appreciate Reviewer 3 for their insightful inquiry. We apologize for any confusion regarding the concept of "measured precision." We recognize the need for clarity in this aspect.

Within Supplementary Section 7: PIC finite precision analysis and its effect on RL, we extensively discuss the theoretical sources of error inherent to the HyArch PIC model. The term "measured precision" pertains to the discrepancy between the calibrated optical dot product outcome and the corresponding theoretical dot product result. Essentially, it serves as a representation of the comprehensive precision achieved within

the PIC calculation system. This metric effectively quantifies the accuracy performance of the PIC system's encoded calculations.

We value Reviewer 3's input, as it has prompted us to provide additional elucidation on the "measured precision" concept. We also want to highlight that Fig. 3F within the manuscript effectively showcases the experimental outcomes of this "measured precision," providing a tangible representation of the precision achieved through our PIC-based calculations.

Figure 3: OCTOPUS module calibration and optical dot product test. (F) Residuals of the optical dot product operation calibrated by unit (gray) and by link (green).

Comment 5: - Using thermo-optic phase shifters results in slower modulation and higher power consumption (80 mW per phase shifter). Please discuss potential ways to overcome the limitations in phase shifter modulation speed and power consumption. Please see the next point.

We extend our gratitude to Reviewer 3 for their insightful observations. The utilization of thermo-optic phase shifters does indeed come with inherent challenges, such as slower modulation speeds and higher power consumption (around 80 mW per phase shifter). While acknowledging these limitations, it's important to emphasize that the use of a network of thermo-optic phase shifters has been widely employed for fundamental

demonstrations and has garnered recognition within the academic community.

Addressing the concerns regarding phase shifter modulation speed and power consumption, there are strategies available to mitigate these limitations. Techniques like designing thermal insulation slots and paperclip waveguides offer avenues to reduce power consumption. Our future efforts will continue to refine MZI designs, with a focus on minimizing modulation power consumption.

The modulation speed of thermo-optic modulators is intrinsically constrained by their underlying modulation mechanism, typically limited to rates below MHz. To achieve significantly higher modulation speeds on the order of GHz, the exploration of electro-optic modulators (EOM) founded on the electro-optic effect becomes pivotal. Introducing faster modulators into the network represents a crucial direction for our ongoing research.

Comment 6: - (major) The high-efficiency claim seems to be based on the assumption that all the MZIs will implement electro-optic modulators demonstrated in another work. However, the authors did not show any experimental data proving such a proposed solution can be implemented in their current design. Operating a single MZI at 100 GHz may be feasible, but operating all MZIs in all meshes at 100 GHz would be a far fetch. There will be many limiting factors, and the estimated speed and energy consumption are significantly over-optimistic. To back up this claim, the authors should demonstrate experimental data from a device that has implemented either an MZI mesh with either EOMs or MEMSs.

We appreciate Reviewer 3 for raising the question regarding our high-efficiency claim. To clarify, our assertion of high efficiency primarily relates to the advantage of our PIC-based SRF RL algorithm over the CRF RL algorithm. While replacing thermal phase modulators with Electro-Optic Modulators (EOMs) or Micro-Electro-Mechanical Systems (MEMS) holds promise for enhancing computational power, it falls into the broader context of 'superiority of photonic computing.'

It's essential to note that the implementation of our hyper HyArch PIC, as described in Supplementary Section 8, doesn't require replacing all MZI structures with EOM or MEMS devices. Only the first column of MZIs in each OCTOPUS module necessitates such replacement. However, developing an EOM or MEMS-based photonic network for reliable operation, capable of handling multiple channels at gigahertz speeds, involves significant engineering challenges beyond the scope of this manuscript.

Our primary focus in this paper is to showcase the efficiency of our algorithm and demonstrate complex RL algorithms on PICs. This is supported by experimental data, as illustrated in Fig 4.E and 5.D of our manuscript, which respectively demonstrate efficiency improvements of 30.6% and 56.25%. These figures provide compelling evidence that underpins our claims of high efficiency.

Based on this valuable comment, we thoroughly reviewed our calculation model and implemented significant modifications. Given the current landscape in scientific research and industry, achieving 100GHz programmable optical networks appears to be highly ambitious. However, programmable optical networks within the range of 100MHz to 10GHz are generally considered feasible, with several demonstrative works providing specific technical insights. In addition to increasing the system frequency of individual chips, the utilization of inter-chip connections and various multiplexing technologies is widely adopted to enhance the computational power of optoelectronic computing architectures. In our updated supplementary material, we have referenced recent breakthroughs in photonic computing chips. We trust that these references, combined with the refinements made to our model, will facilitate a more comprehensive evaluation of high-speed photonic computing chips. Below is our modification to ‘Supplementary Section 8: Computational power and energy efficiency analysis’:

“It should be noted that the bandwidth of a single EOM modulator can exceed 100GHz [18], but the system frequency of the EOM-based PIC network may be constrained by the multi-channel electrical driver. In our comprehensive feasibility assessment [21-25], we conducted calculations to evaluate the computational power of the hyper-HyArch PIC system at three distinct operating frequencies: 100MHz, 1GHz, and 10GHz. These results were then compared to the computational power of corresponding electrical hardware and artificial intelligence models. Our HyArch PIC architecture exhibits high scalability, allowing for significant expansion by increasing the size of the monolithic network [25] and employing inter-chip optical interconnect technology [26,27]. Fig. S12 illustrates the relationship between computing power and network size (M), revealing that the computing power of HyArch PIC grows quadratically with M . As depicted in Fig. S12, even with a 100MHz electric drive system, the computing power offered by a relatively small-scale HyArch PIC system is sufficient to meet the computational demands of practical neural networks like AlexNet and ResNet-152 [28]. Furthermore, when operating in synergy with 10GHz drivers, 128-dimensional HyArch PIC systems can provide computing power comparable to that of the A100, which serves as a primary source of computational capability in modern data centers.

In the advanced PIC process, the power consumption of the optical components is nearly negligible when compared to the high-energy demands of the high-speed electrical drive. Below we mainly calculate the power consumption of the electrical part. We simulated the lithium niobate on insulator (LNOI) EOM model through finite element analysis and calculated that the equivalent capacitance of 5mm LNOI EOM is 0.3pF to estimate the power consumption of EOM. Assuming a 2V drive voltage at 1GHz, the power consumption of EOM is $P_{EOM} = CV^2\Delta f_{EOM} = 1.2mW$. At 1GHz system frequency, the power consumption of DAC/ADC/TIA is 11.06mW, 2.55mW and 57mW respectively.

Theoretical energy efficiency calculation result for a 128-dimensional HyArch PIC operating at 1GHz is 5.86pJ/OP, and most of the energy consumption comes from the high-speed DAC (94.45%). Further reducing the power consumption of the DAC can bring the energy efficiency of the whole system close to the commercial GPU.”

Table S3: Summary of parameters in numerical calculations

Category		value	ref
Capacity of EOM		0.3 pF	Simulated
DAC (transmitter, 1GHz)		11.06mW	[29]
ADC (receiver, 1GHz)		2.55mW	[30]
TIA (receiver, 1GHz)		57mW	[31]
MEMS (weight, stable)		75 μ W	[19]
Computating power	JETSON TX1	1 TOPS	[32]
	RTX 2080	89.2 TOPS	[33]
	A100	312 TOPS	[34]
Power consumption	JETSON TX1	15W	[32]
	RTX 2080	225W	[33]
	A100	400W	[34]

Fig. S12: The computational power of hyper-HyArch PIC as a function of module number M at different system frequencies.

- [21] Ying, Z. et al *Electronic-photonic arithmetic logic unit for high-speed computing*. *Nature communications* 11, 2154 (2020).
- [22] Dong, M. et al *High-speed programmable photonic circuits in a cryogenically compatible, visible–near-infrared 200 nm CMOS architecture*. *Nature Photonics* 16, 59–65 (2022).
- [23] Mourgias-Alexandris, G. et al *Noise-resilient and high-speed deep learning with coherent silicon photonics*. *Nature Communications* 13, 5572 (2022).
- [24] Lin, Z. et al *High-performance polarization management devices based on thin-film lithium niobate*. *Light: Science & Applications* 11, 93 (2022).
- [25] Peng, B., Hua, S., Su, Z., Xu, Y. & Shen, Y. *A 64× 64 integrated photonic accelerator*. In *2022 IEEE Photonics Conference (IPC)*, 1–2 (IEEE, 2022).
- [26] Abrams, N. C. et al *Silicon photonic 2.5 d multi-chip module transceiver for high performance data centers*. *Journal of Lightwave Technology* 38, 3346–3357 (2020).
- [27] Hummingbird: Lightelligence optical network-on-chip accelerator. <https://www.lightelligence.ai/index.php/product/hummingbird.html>
- [28] Zhang, M. et al *Optimized compression for implementing convolutional neural networks on FPGA*. *Electronics* 8, 295 (2019).
- [29] Demirkiran, C. et al *An electro-photonic system for accelerating deep neural networks*. *arXiv preprint arXiv:2109.01126* (2021).
- [30] Oh, D.-R. et al *An 8b 1GS/s 2.55 mW SAR-flash ADC with complementary dynamic amplifiers*. In *2020 IEEE Symposium on VLSI Circuits*, 1–2 (IEEE, 2020).
- [31] Sedighi, B. & Scheytt, J. C. *Low-power SiGe BiCMOS transimpedance amplifier for 25-GBaud optical links*. *IEEE Transactions on Circuits and Systems II: Express Briefs* 59, 461–465 (2012).

Correspondingly, we have also made the following modifications to the discussion section of the manuscript:

“By optimizing the thermal optical modulator in the HyArch PIC with EOM and MEMS modulators, and considering a configuration with $M = 128$, at a system frequency of $f_s = 10\text{GHz}$, the number of operations per second (OPS), denoted as $R \sim 2Nf_s = 2M^2f_s$, for the HyArch PIC will be comparable to that of the state-of-the-art GPU (NVIDIA A100). The energy consumption analysis of the HyArch PIC suggests that most of the energy is consumed in high-speed digital-to-analog conversion (DAC). Reducing this part of energy consumption is necessary for the development of future optoelectronic hybrid computing architecture (see section S8). The optoelectronic hybrid computing architecture, recognized for its notable scalability and robust computing capabilities, stands as a compelling contender for the next-generation AI hardware platform, especially when integrated heterogeneously with AI-specific hardware [60, 61], which further enhances its performance.”

We would like to express our gratitude once again for the constructive comment from Reviewer 3. Their comments not only provided us with an opportunity to elucidate the concept of high-efficiency but also aided us in refining and revising our claims regarding the model of HyArch PIC computational power.

Comment 7: - References: Some missing references to recent developments

- This work does not have an integrated nonlinear activation function critical for an optical neural network. The authors discuss quadratic nonlinearity that was first demonstrated 10.1103/PhysRevApplied.11.064043 as a programmable photonic modulator neuron which is overlooked.

- Authors should also consider referencing Bandyopadhyay, Saumil, et al. "Single chip photonic deep neural network with accelerated training." arXiv preprint arXiv:2208.01623 (2022).

- There has been much work on optical Ising machines and optical neural networks. Consider adding recent references to review articles for the readers:

Mohseni, N., McMahon, P.L. & Byrnes, T. Ising machines as hardware solvers of combinatorial optimization problems. Nat Rev Phys 4, 363–379 (2022).

Shastri, B.J., Tait, A.N., Ferreira de Lima, T. et al. Photonics for artificial intelligence and neuromorphic computing. Nat. Photonics 15, 102–114 (2021).

Huang, C., Fujisawa, S., de Lima, T.F. et al. A silicon photonic–electronic neural network for fibre nonlinearity compensation. Nat Electron 4, 837–844 (2021).

We sincerely appreciate Reviewer 3 for highlighting these relevant and recent developments in the field of optical neural networks. Your insightful suggestions are invaluable, and we will ensure to incorporate the missing references into our manuscript as appropriate. The works you've pointed out, including the demonstration of quadratic nonlinearity as a programmable photonic modulator neuron (10.1103/PhysRevApplied.11.064043), the single-chip photonic deep neural network with accelerated training (arXiv:2208.01623), and the comprehensive review articles on Ising machines and photonics for artificial intelligence and neuromorphic computing, provide valuable context and depth to our research. We are committed to enriching our manuscript by incorporating these references and acknowledging their significance in the broader landscape of optical neural networks and related technologies.

REVIEWER COMMENTS

Reviewer #1 (Remarks to the Author):

REVISIONS BASED ON MY INITIAL REVIEW ARE SATISFACTORY

Reviewer #2 (Remarks to the Author):

After a comprehensive evaluation of the manuscript and careful consideration of its suitability for publication in high impact journal of Nature Communications, I still cannot recommend it for publication due to its lack of novelty and contributions:

1. The photonic integrated circuits presented in this work essentially execute the dot product/matrix multiplication and accumulation, and are still designed based on simple MZIs, which exhibit very limited contributions from the perspective of developing photonic integrated circuits. There already have been many publications in this area, even the universal linear optical circuit of MZIs that can accomplish any unitary matrix has reported a matrix dimension of 15-by-15, ~210 reconfigurable parameters (G.S. Thekkadath et al, PRX Quantum 3, 020336, 2022). Compared to this work, the results reported in this work for U (3) and 87 parameters are somehow out-of-date.

2. The manuscript has limited novelty and contributions in the photonic integrated circuit part; the only interesting feature is the application scenario of reinforcement learning. However, I cannot agree that it is an efficient design for reinforcement learning, especially the claim that "has the potential to outperform the existing electronic computing architectures in computing power performance" due to the following reasons. The reason why CPU is falling short of computation power for NNs has two main reasons: the computation burden of enormous MACs, and the repeated retrieval and deployment of intermediate results of NNs from memory. In reinforcement learning, the latter problem is especially severe since the agent needs extensive experience in interacting with the environment. In this demonstration, it would require enormous optical-electrical conversion. Using thermo-optical phase shifters, the modulation frequency has an upper bound of tens KHz, making it uncompetitive compared with electronic devices. For the MACs, the paper also has limited novelty by using simple MZIs, as said in comment 1.

Considering its limited contributions to photonic integrated circuit dot products and its limited advantages in reinforcement learning, I cannot accept it for publication.

Reviewer #3 (Remarks to the Author):

I thank the authors for taking my comments seriously and addressing them to the best of their ability. While all technical challenges cannot be solved in a manuscript, I appreciate the authors' attempt and insight in recognizing and providing plausible explanations of those challenges and how they could potentially be solved. Overall, I am satisfied with their response to most of my concerns; in particular, they have provided nice explanations to the questions on the high-efficiency claim (comment 6), thermo-optic phase shifters (comment 5), and the bit precision claim (comment 4). However, the novelty of the hardware is not coming through in the manuscript. I still find the work impressive, but I would recommend that the authors explicitly state the architectural innovations. The authors should comment on and acknowledge photonic reinforcement learning (based on reservoir computing approaches) approaches in literature:

J. Bueno, S. Maktoobi, L. Froehly, I. Fischer, M. Jacquot, L. Larger, and D. Brunner, "Reinforcement learning in a large-scale photonic recurrent neural network," *Optica* 5, 756-760 (2018)

Kanno, K., Uchida, A. Photonic reinforcement learning based on optoelectronic reservoir computing. *Sci Rep* 12, 3720 (2022). <https://doi.org/10.1038/s41598-022-07404-z>

Re: NCOMMS-23-25469A
Response to Reviewers' Report

We would like to express our sincere gratitude once again for the dedicated efforts of the editors and reviewers in their thorough assessment of our manuscript, titled "High-efficiency Reinforcement Learning with Hybrid Architecture Photonic Integrated Circuit." We deeply appreciate the editors and reviewers for affording us a second opportunity to revise our manuscript, as well as for providing invaluable comments and suggestions that significantly enriched the depth and comprehensiveness of our work.

We summarize the whole picture for the last two rounds:

- 1) Reviewer 1 expressed satisfaction with our initial revisions and holds a positive view of our work.
- 2) Reviewer 3 also found our first round of revisions generally satisfactory and had a positive opinion of our work. They suggested enhancing the descriptions of hardware superiority and algorithm innovation.
- 3) Reviewer 2 considers that the contribution of our work is limited and holds a critical perspective.
- 4) The editor has requested that we address the hardware and algorithm innovation concerns raised by Reviewers 2 and 3 to facilitate a more comprehensive evaluation of our work.

We have addressed all the comments and suggestions in the Response, and added more corresponding detailed descriptions in the revised manuscript (marked in blue). We are confident that the revised manuscript is now suitable for publication in *Nature Communications*.

Brief Summary of Changes:

We have incorporated these suggestions into our revised manuscript, which at a high level, included the following changes:

- The first paragraph in the Discussion section has been extensively revised to offer a thorough and explicit clarification of the architectural advancement.
- The third paragraph in the Discussion section has been rephrased to explore potential approaches for further integrating RL and PIC,

encompassing the methodology proposed by Reviewer 3 involving reservoir computing.

- The figure label's capitalization has been modified to adhere to the formatting standards of *Nature Communications*.
- The sections 'Acknowledgments,' 'Author contributions,' 'Competing interests,' and 'Code availability' have been adjusted to comply with the formatting standards of *Nature Communications*.

Specific responses to the reviewer comments are given below, **with our answers written in blue**. Changes to the main text have also been highlighted in the provided PDF.

The Authors

Response to the Report of Reviewer #1

REVISIONS BASED ON MY INITIAL REVIEW ARE SATISFACTORY

We genuinely thank Reviewer 1 for their positive feedback on the revisions made in response to their initial review. Their constructive comments in the first round of review were instrumental in elevating the quality and depth of our manuscript, and we are highly appreciative of their valuable contributions.

Response to the Report of Reviewer #2

After a comprehensive evaluation of the manuscript and careful consideration of its suitability for publication in high impact journal of Nature Communications, I still cannot recommend it for publication due to its lack of novelty and contributions:

- 1. The photonic integrated circuits presented in this work essentially execute the dot product/matrix multiplication and accumulation, and are still designed based on simple MZIs, which exhibit very limited contributions from the perspective of developing photonic integrated circuits.*

There already have been many publications in this area, even the universal linear optical circuit of MZIs that can accomplish any unitary matrix has reported a matrix dimension of 15-by-15, ~210 reconfigurable parameters (G.S. Thekkadath et al, PRX Quantum 3, 020336, 2022). Compared to this work, the results reported in this work for $U(3)$ and 87 parameters are somehow out-of-date.

The reference mentioned by Reviewer 2 relates to photonic GBS, in which the optical chip is entirely passive, without an electric driver to modulate phase parameters within the chip. The original text describing the chip in the article reads as follows:

“The reflectivities are chosen to follow a Haar-random distribution while the phases are randomised due to the fabrication tolerance [14]. Since we fix the three input modes in our experiment, we only characterize the relevant 3×15 submatrix.”

From the original text, we can see that this experiment solely utilized a 3×15 Haar random matrix supplied by the passive optical chip. Therefore Reviewer 2's claim of '~210 reconfigurable parameters' for this work is completely inappropriate.

Regarding Reviewer 2's remark that "the results reported in this work for U(3) and 87 parameters are somehow out-of-date," we respectfully disagree with this viewpoint. The 87 on-chip adjustable parameters correspond to a 9*9 fully programmable MZI mesh network. Photonic hardware of this scale is still the mainstream configuration for demonstrating photonic AI algorithms. In the field of optical linear neuron architectures, our manuscript achieves high-precision linear dot product operations in dimensions as high as 15, constituting the most significant parallel experimental demonstration of such operations on an optical linear neuron chip to our knowledge. This significant advancement stems from the architectural advantages of the HyArch PIC and the precision enhancements facilitated by our unique link calibration methodology.

We have modified the first paragraph of the Discussion to fully discuss the innovation and advancement of the hardware architecture as following:

“HyArch PIC integrates unitary MZI mesh architecture and coherent linear neuron architecture, such as OCTOPUS, into a monolithic PIC framework, significantly enhancing its capabilities. This hybrid architecture offers several distinct advantages over standalone unitary MZI mesh and coherent linear neuron architectures, including: 1) exceptional scalability and robust fault tolerance, 2) versatile functionality and 3) high-speed compatibility. We employ cosine distance \mathcal{D} to assess the scalability and fault tolerance of the PIC architecture. Finite precision analysis reveals that the N -dimensional HyArch PIC and SVD mesh architectures exhibit cosine distances $\mathcal{D}_{H(N)} \sim 2\sqrt{N} \log(N) \sigma_{BS}^2$ and $\mathcal{D}_{SVD} \sim 4N \sigma_{BS}^2$, respectively (see section S7). This indicates that HyArch PIC has a sub-exponential advantage over SVD architecture PIC in terms of scalability and fault tolerance. The overall transmission matrix of the N -dimensional HyArch PIC $\mathbf{T}_{HyArch\ PIC}$ (composed of an M -dimensional U mesh and M OCTOPUS modules, where $N = M^2$) can be expressed as follows:

$$\mathbf{T}_{HyArch\ PIC} = \mathbf{T}_{U(M)} \mathbf{T}_{O_M} = \mathbf{W}_{M \times M} \begin{bmatrix} \mathbf{u}_1 \cdot \mathbf{v}_1 \\ \vdots \\ \mathbf{u}_M \cdot \mathbf{v}_M \end{bmatrix} = \mathbf{W}_{M \times M} \begin{bmatrix} \sum_{i=1}^M u_{1_i} v_{1_i} \\ \vdots \\ \sum_{i=1}^M u_{M_i} v_{M_i} \end{bmatrix}$$

where $\mathbf{W}_{M \times M}$ is the weight matrix provided by the front U module, and the M OCTOPUS modules perform M sets of M -dimensional dot product operations $\mathbf{u}_m \cdot \mathbf{v}_m, m = 1, 2, \dots, M$. In terms of functionality, HyArch PIC can deploy weighted group dot product/MVM, a crucial component in advanced ML algorithms such as the weighted multi-core convolution for computer vision [56], multi-head attention in natural language processing [5], and more. The innovative HyArch PIC significantly

enhances the expressive power of PIC hardware, enabling its broader applicability in machine learning and distributed computing. Elevating the speed of electro-optical modulation in PIC hinges on key factors: channels for high-speed electric drive and DAC. These factors directly determine the system frequency for the entire chip. In N-dimensional optical dot product tasks, HyArch PIC stands out, requiring only approximately 1/N of the modulation units compared to MZI mesh architecture (see section S7). This significant reduction simplifies integration with high-speed electric drive, bringing optoelectronic computing chips closer to contemporary commercial GPUs in computing power. What's more, HyArch PIC offers exceptional modularity, delivering a more adaptable layout compared to the one-way expansion MZI mesh and OCTOPUS architectures. This adaptability makes it particularly well-suited for high-density RF photonic packaging.”

[5] Vaswani, A. et al Attention is all you need. Advances in neural information processing systems 30 (2017).

[56] Cho, W., Son, S. & Kim, D.-S. Weighted multi-kernel prediction network for burst image super-resolution. In Proceedings of the IEEE/CVF Conference on Computer Vision and Pattern Recognition, 404–413 (2021).

2. The manuscript has limited novelty and contributions in the photonic integrated circuit part; the only interesting feature is the application scenario of reinforcement learning. However, I cannot agree that it is an efficient design for reinforcement learning, especially the claim that “has the potential to outperform the existing electronic computing architectures in computing power performance” due to the following reasons. The reason why CPU is falling short of computation power for NNs has two main reasons: the computation burden of enormous MACs, and the repeated retrieval and deployment of intermediate results of NNs from memory. In reinforcement learning, the latter problem is especially severe since the agent needs extensive experience in interacting with the environment. In this demonstration, it would require enormous optical-electrical conversion. Using thermo-optical phase shifters, the modulation frequency has an upper bound of tens KHz, making it uncompetitive compared with electronic devices. For the MACs, the paper also has limited novelty by using simple MZIs, as said in comment 1.

While the full potential has not been fully realized currently, it is widely acknowledged in both academic and industrial circles that optoelectronic integrated computing systems have the capacity to accelerate large-scale neural network calculations. Compared to electronic hardware like CPUs, which rely on bitwise multiplication and

addition operations to perform dot products and matrix-vector multiplications (MVM), PICs leverage their inherent high degree of parallelism to accelerate these operations, resulting in a significant boost in computing power.

In our manuscript, we emphasize the significant advantages of using PIC technology in RL applications. The conventional challenge in RL is the extensive experience required for the agent to interact with the environment effectively. However, it's important to note that the off-policy nature of Q-learning provides a solution to this problem by allowing the agent to learn from historical data efficiently, reducing the demand for real-time interaction. In our PIC-RL scheme, the reward only needs to be calculated globally once using PIC, and the generated reward table is stored in the electronic device for subsequent retrieval, which keeps the optical-electrical conversion manageable. Additionally, whether it involves high-dimensional similarity calculations in the reward function, as mentioned in our manuscript, or large-scale neural network calculations as seen in DQN, these operations are essentially high-dimensional dot product and matrix-vector multiplication (MVM). The PIC technology has the potential to accelerate these operations, further enhancing the efficiency of RL algorithms.

In conclusion, while the need for extensive experience in RL is a recognized challenge, the combination of off-policy Q-learning and PIC technology in our approach mitigates this issue. The PIC-RL process, with its efficient R-table calculation and accelerated operations, holds the promise of advancing RL in various applications, as indicated by the substantial performance improvements in our experiments.

In the Discussion section, we have envisioned further potential applications of RL in the field of photonic integrated circuits:

“The introduction of cosine similarity into the reward function highlights its effectiveness in training RL models within finite discrete environmental spaces. Furthermore, the technology of inverse reinforcement learning (IRL) can be used to further optimize the reward function to improve algorithm efficiency, which is also one of the key directions of research in the RL domain. Our research is centered on leveraging PIC to address RL tasks in finite discrete state/action space. Given the limited storage capacity of a Q-table, Q-learning proves to be an effective approach for solving such tasks. When addressing RL tasks with high-dimensional or nearly continuous state/action spaces, Deep Q-Network (DQN) [4,62] utilizes a neural network to approximate the Q-function, effectively replacing the traditional Q-table. Optoelectronic cointegration technology such as on-chip electrical logic and nonlinear units will greatly enrich the functions of photonic computing architecture [23,26,63,64],

which will promote the further development of PIC-RL, and make PIC-based DQN possible (see section S9). By combining with reservoir computing, the model parameters required to build DQN can be significantly reduced, which also offers an efficient approach for transitioning DQN into optoelectronic co-integration systems [65].”

Our work originally introduced a photonic hardware architecture, thoroughly characterized its advanced performance through experiments, extended its application to new fields, and successfully yielded significant results. We consider this process to be comprehensive enough for an original and inspiring research, with the expansion of PIC dimensionality and improvements in computing power falling well within the technical scope of future research. And we believe that with the continuous development of optoelectronic integration technology, such as system on a chip (SoC) and co-packaged optics (CPO), the energy consumption of optical-electrical conversion will continue to decrease in the future.

Considering its limited contributions to photonic integrated circuit dot products and its limited advantages in reinforcement learning, I cannot accept it for publication.

We appreciate the feedback provided by Reviewer 2 and would like to express our gratitude for taking the time to evaluate our manuscript. While we understand the concerns raised, we kindly request that Reviewer 2 reconsider their evaluation from a more objective, equitable, and comprehensive standpoint. We have compared our work to the recent representative works in integrated optical neural networks over the past few years. Our work demonstrates a sufficient level of advancement and innovation in terms of hardware innovation, comprehensive calibration, encoding precision, encoding dimensionality, and innovative applications.

Table1. Comparison of Different Integrated Optical Neural Network Work

Representative works	PIC architecture	Real time deterministic encoding	Residual error	Encoding vector dimensions	Application
Nat. Photon. 11, 441–446 [1]	Unitary MZI mesh	NO	0.0224	4	Vowel recognition (interference)

Nat. Commun. 12, 457 [2]	Unitary MZI mesh	NO	/	4	Iris/MNIST classification
Light Sci. Appl. 10, 221 [3]	Linear neuron	NO	0.0104	3	Regression tasks
Nat. Commun. 13, 5572 [4]	Linear neuron	NO	0.21	2	MNIST classification
Our work	Hybrid architecture	YES	0.0166	14	Q-learning training (RL)

- [1] Shen, Yichen, et al. "Deep learning with coherent nanophotonic circuits." *Nature photonics* 11.7 (2017): 441-446.
- [2] Zhang, Hui, et al. "An optical neural chip for implementing complex-valued neural network." *Nature communications* 12.1 (2021): 457.
- [3] Xu, Shaofu, et al. "Optical coherent dot-product chip for sophisticated deep learning regression." *Light: Science & Applications* 10.1 (2021): 221.
- [4] Mourgias-Alexandris, G., et al. "Noise-resilient and high-speed deep learning with coherent silicon photonics." *Nature Communications* 13.1 (2022): 5572.

We firmly assert that our manuscript carries meaningful contributions to the field of photonic integrated hardware. The innovative design of the HyArch PIC architecture, for instance, enables high-dimensional parallel optical on-chip dot products, which holds substantial significance in the advancement of photonic integrated circuits. Additionally, the link calibration method we introduce has the potential to markedly enhance the accuracy of on-chip dot products, thus adding to the relevance and utility of our work.

Furthermore, our manuscript demonstrates the successful application of this hardware in addressing complex RL tasks within a high dimensional state space. We are confident that this accomplishment not only highlights the effectiveness of our hardware design but also holds broader significance for the integrated photonics community. The methods and experimental insights outlined in our manuscript can provide valuable guidance and serve as a reference for researchers and practitioners in the field.

Response to the Report of Reviewer #3

I thank the authors for taking my comments seriously and addressing them to the best of their ability. While all technical challenges cannot be solved in a manuscript, I appreciate the authors' attempt and insight in recognizing and providing plausible explanations of those challenges and how they could potentially be solved. Overall, I am satisfied with their response to most of my concerns; in particular, they have provided nice explanations to the questions on the high-efficiency claim (comment 6), thermo-optic phase shifters (comment 5), and the bit precision claim (comment 4).

We genuinely appreciate Reviewer 3 for their positive feedback on the revisions made in response to their initial review. Their constructive comments and suggestions have greatly improved the quality of our article.

However, the novelty of the hardware is not coming through in the manuscript. I still find the work impressive, but I would recommend that the authors explicitly state the architectural innovations.

We are very grateful to Reviewer 3 for the suggestions on explicitly demonstrating the innovativeness of the hardware architecture. We have modified the first paragraph of Discussion to fully discuss the innovation and superiority of HyArch PIC. The revised manuscript is as follows:

“HyArch PIC integrates unitary MZI mesh architecture and coherent linear neuron architecture, such as OCTOPUS, into a monolithic PIC framework, significantly enhancing its capabilities. This hybrid architecture offers several distinct advantages over standalone unitary MZI mesh and coherent linear neuron architectures, including: 1) exceptional scalability and robust fault tolerance, 2) versatile functionality and 3) high-speed compatibility. We employ cosine distance \mathcal{D} to assess the scalability and fault tolerance of the PIC architecture. Finite precision analysis reveals that the N -dimensional HyArch PIC and SVD mesh architectures exhibit cosine distances $\mathcal{D}_{H(N)} \sim 2\sqrt{N} \log(N) \sigma_{BS}^2$ and $\mathcal{D}_{SVD} \sim 4N \sigma_{BS}^2$, respectively (see section S7). This indicates that HyArch PIC has a sub-exponential advantage over SVD architecture PIC in terms of scalability and fault tolerance. The overall transmission matrix of the N -dimensional HyArch PIC $\mathbf{T}_{HyArch\ PIC}$ (composed of an M -dimensional U mesh and M OCTOPUS modules, where $N = M^2$) can be expressed as follows:

$$\mathbf{T}_{HyArch\ PIC} = \mathbf{T}_{U(M)} \mathbf{T}_{O_M} = \mathbf{W}_{M \times M} \begin{bmatrix} \mathbf{u}_1 \cdot \mathbf{v}_1 \\ \vdots \\ \mathbf{u}_M \cdot \mathbf{v}_M \end{bmatrix} = \mathbf{W}_{M \times M} \begin{bmatrix} \sum_{i=1}^M u_{1_i} v_{1_i} \\ \vdots \\ \sum_{i=1}^M u_{M_i} v_{M_i} \end{bmatrix}$$

where $\mathbf{W}_{M \times M}$ is the weight matrix provided by the front U module, and the M OCTOPUS modules perform M sets of M -dimensional dot product operations $\mathbf{u}_m \cdot \mathbf{v}_m, m = 1, 2, \dots, M$. In terms of functionality, HyArch PIC can deploy weighted group dot product/MVM, a crucial component in advanced ML algorithms such as the weighted multi-core convolution for computer vision [56], multi-head attention in natural language processing [5], and more. The innovative HyArch PIC significantly enhances the expressive power of PIC hardware, enabling its broader applicability in machine learning and distributed computing. Elevating the speed of electro-optical modulation in PIC hinges on key factors: channels for high-speed electric drive and DAC. These factors directly determine the system frequency for the entire chip. In N -dimensional optical dot product tasks, HyArch PIC stands out, requiring only approximately $1/N$ of the modulation units compared to MZI mesh architecture (see section S7). This significant reduction simplifies integration with high-speed electric drive, bringing optoelectronic computing chips closer to contemporary commercial GPUs in computing power. What's more, HyArch PIC offers exceptional modularity, delivering a more adaptable layout compared to the one-way expansion MZI mesh and OCTOPUS architectures. This adaptability makes it particularly well-suited for high-density RF photonic packaging.”

[5] Vaswani, A. et al Attention is all you need. Advances in neural information processing systems 30 (2017).

[56] Cho, W., Son, S. & Kim, D.-S. Weighted multi-kernel prediction network for burst image super-resolution. In Proceedings of the IEEE/CVF Conference on Computer Vision and Pattern Recognition, 404–413 (2021).

We hope that the above discussion and the theoretical analysis in section S7 of the supplementary material can fully and intuitively demonstrate the innovation and superiority of HyArch PIC.

The authors should comment on and acknowledge photonic reinforcement learning (based on reservoir computing approaches) approaches in literature:

Brunner, "Reinforcement learning in a large-scale photonic recurrent neural network," Optica 5, 756-760 (2018)

In this article, the authors have developed a sophisticated free-space optical system to showcase a large-scale photonic recurrent neural network. The primary advantage lies in the system's capacity to attain an extensive network scale, endowing the photonic RNN with substantial expressive capabilities. However, it's important to note that due to its free-space nature, this system lacks the integration level achievable with our PIC system. Furthermore, it exhibits sensitivity to environmental disturbances and relatively poor robustness. Significantly, the use of Spatial Light Modulators (SLM) and Digital Micromirror Devices (DMD) as modulation units imposes constraints on both modulation speed and accuracy. The paper notes a refresh rate of 5Hz and confines the operations to Boolean functions, highlighting a critical trade-off in these dimensions.

Regarding the RL aspect of the article, the author did not provide clarity on the state/action space or the specific RL model employed. Instead, the focus was primarily on the construction of the reward function and the weight update process. Hence, it can be inferred that the primary purpose of RL in this article is to enhance the performance of the free-space RNN system (or the reservoir computing system as discussed in the article). By employing the reward function construction and weight update method, RL assumes a pivotal role in optimizing the RNN, thereby substantiating our assertion regarding the critical importance of reward function construction.

$$r(k) = \begin{cases} 1, & \text{if } \varepsilon_k < \varepsilon_{k-1} \\ 0, & \text{if } \varepsilon_k \geq \varepsilon_{k-1} \end{cases}$$

$$W_{l_k, k}^{DMD} = r(k)W_{l_k, k}^{DMD} + (r(k) - 1)W_{l_k, k-1}^{DMD}$$

In summary, this article uses RL as a technology to optimize the encoding matrix of DMD in the optical path, and achieves effective training effects by constructing a specific reward function. In comparison to our manuscript, where we utilize a PIC to simulate the core interactions between the agent and environment in RL and perform extensive computations, the application of RL in this work leans more towards technical and optimization aspects rather than presenting innovative principles. Similarly, we believe that RL, as a highly feasible machine learning algorithm, holds extensive potential applications in areas such as optical system parameter tuning, optical component optimization, and the automated design of optical neural networks.

Kanno, K., Uchida, A. Photonic reinforcement learning based on optoelectronic reservoir computing. Sci Rep 12, 3720 (2022).

We greatly appreciate Reviewer 3 for providing this relevant work. While the device

and implementation method differ significantly, this work can be considered as a valuable complement to our own research. In the discussion section, we highlighted the suitability of DQN for addressing challenges in larger, nearly continuous state/action spaces. This reference demonstrates the use of reservoir computing (RC) as a replacement for the neural network component in DQN to represent the Q function. This approach significantly reduces the model's parameter count and facilitates enhanced implementation within the optoelectronic computing architecture. Reservoir computing's output represents the Q value corresponding to the same action in different states. Its expression can be written in the form of matrix-vector multiplication, a task well-suited for optoelectronic computing system:

$$Q(\mathbf{s}_n, a) = \sum_{j=1}^N w_{j,a} v_{j,n} = \mathbf{w}_a^T \mathbf{v}_n$$

The reference literature computes the Q value using a single MZM, achieving limited acceleration. And since only MZM encoding weights are used, the photonics part of the entire framework is relatively limited. Migrating this method similar to that in Section 9 of our supplementary material to the HyArch PIC system can further accelerate the DQN algorithm.

In summary, our work diverges from the referenced study in two significant ways. Firstly, while both researches are focused on enhancing the computational efficiency of the Q-function in RL using optoelectronic architectures, the referenced work adopts a network-based approach, where the Q-function relies on network weights and inputs. Although this neural network calculation is applied to the RL field, essentially the calculation of multiplying weights and inputs is similar to the optical neural network architecture that has already been demonstrated. In contrast, our manuscript focuses on the direct and efficient computation of the Q-function using the HyArch PIC to enhance RL efficiency while circumventing the challenges of migrating large-scale neural networks (such as DQN or RC) to the PIC. The reference indicates that even the relatively modest RC network requires 600 nodes.

Additionally, our integrated optoelectronic computing system, unlike the referenced work's reliance on a single MZM-based system, exhibits higher levels of integration and parallelism. The referenced work limited its optical computing system to weight multiplication, with the majority of computations conducted in the electrical domain. In contrast, our integrated optoelectronic computing system excels at performing precise calculations of the complete R-table within a 3472-dimensional state space, entirely within the optical domain.

Despite these distinctions, we acknowledge the potential for the referenced work to contribute to the convergence of photonic integrated circuits and reinforcement learning. We cited this reference and incorporated the following revisions into our discussion part of RL algorithms:

“The introduction of cosine similarity into the reward function highlights its effectiveness in training RL models within finite discrete environmental spaces. Furthermore, the technology of inverse reinforcement learning (IRL) can be used to further optimize the reward function to improve algorithm efficiency, which is also one of the key directions of research in the RL domain. Our research is centered on leveraging PIC to address RL tasks in finite discrete state/action space. Given the limited storage capacity of a Q-table, Q-learning proves to be an effective approach for solving such tasks. When addressing RL tasks with high-dimensional or nearly continuous state/action spaces, Deep Q-Network (DQN) [4,62] utilizes a neural network to approximate the Q-function, effectively replacing the traditional Q-table. Optoelectronic cointegration technology such as on-chip electrical logic and nonlinear units will greatly enrich the functions of photonic computing architecture [23,26,63,64], which will promote the further development of PIC-RL, and make PIC-based DQN possible (see section S9). By combining with reservoir computing, the model parameters required to build DQN can be significantly reduced, which also offers an efficient approach for transitioning DQN into optoelectronic co-integration systems [65].”

Finally, we would like to thank Reviewer 3 again for the constructive comments on this manuscript, which greatly increased the depth and completeness of our work.

REVIEWERS' COMMENTS

Reviewer #2 (Remarks to the Author):

Although this article presents a significant background of optical neural networks and demonstrates their expertise in the algorithm (reinforcement learning) domain, the study still falls within the scope of demonstrating an algorithm using existing hardware methods. The techniques employed, including the linear optical Mach-Zehnder interferometer (MZI) network and active optoelectronic control, are mature technologies in this field. This work lacks innovation in hardware methods and fails to address these critical challenges in optical neural networks that hinder optical neural networks from truly surpassing their electronic counterparts, such as limited scalability, high photonic conversion costs, and storage requirements, thus lacking a substantial impact on the field. Therefore, I cannot recommend its publication in such a high-impact journal of NC.

Reviewer #3 (Remarks to the Author):

In the first revision, my main concern was that the novelty was not coming through in the manuscript. In the 2nd revision, I am satisfied with the authors' attempt to further clarify the claims and novelty. So, in principle, I would recommend this manuscript for publication. However, all of this discussion is relegated to the discussion section. I would recommend that some of that discussion, especially on the advantages of hybrid architecture (over the standard unitary MZI mesh and coherent linear architectures), be explicitly stated in the introduction.

I thank the authors for considering my suggestions.

Re: NCOMMS-23-25469B
Response to Reviewers' Report

We wish to extend our sincere gratitude to the editors and reviewers for their dedicated efforts in thoroughly evaluating our manuscript, titled "High-efficiency Reinforcement Learning with Hybrid Architecture Photonic Integrated Circuit." Their invaluable comments and suggestions have significantly enhanced the depth and comprehensiveness of our work.

Brief Summary of Changes:

We have incorporated these suggestions into our revised manuscript, which at a high level, included the following changes:

- According to the recommendations from Reviewer 3, we have incorporated the advantages of hybrid architecture explicitly into the introduction section.
- In order to enhance readability, we have transferred some of the excessively detailed hardware descriptions from the main body to the methods section.
- We have confirmed that all supplementary materials are appropriately cited in the main text.
- We have normalized all error distributions and replaced the term "count" with "probability" throughout the manuscript.
- We have refined the expression of the SRF in Equation 2 and provided additional explanations for clarity in depicting our model. Annotations in Fig. 4 and Fig. S7 have been modified. We have replaced the previous fidelity with the standard deviation of the error distribution to describe the accuracy of the experimental calculation results in cliff walking.
- We have made updates to both the data availability and code availability sections.
- We have made typo corrections to the timing of the rising and falling edges of the thermal modulator.

Specific responses to the reviewer comments are given below, with our answers written in blue. Changes to the main text have also been highlighted in the provided PDF.

The Authors

Response to the Report of Reviewer #2

Reviewer #2 (Remarks to the Author):

Although this article presents a significant background of optical neural networks and demonstrates their expertise in the algorithm (reinforcement learning) domain, the study still falls within the scope of demonstrating an algorithm using existing hardware methods. The techniques employed, including the linear optical Mach-Zehnder interferometer (MZI) network and active optoelectronic control, are mature technologies in this field. This work lacks innovation in hardware methods and fails to address these critical challenges in optical neural networks that hinder optical neural networks from truly surpassing their electronic counterparts, such as limited scalability, high photonic conversion costs, and storage requirements, thus lacking a substantial impact on the field. Therefore, I cannot recommend its publication in such a high-impact journal of NC.

First and foremost, we express our sincere gratitude to Reviewer 2 for their continuous engagement and insightful feedback. We particularly value the detailed technical discussions and algorithm-related insights shared during the initial review, which significantly contributed to enhancing the overall depth and clarity of our manuscript. While the subsequent rounds shifted towards more conceptual discussions, this broader perspective has further refined the excellence and innovation inherent in our work. We think the discussion with reviewer 2 elevating the overall quality of our manuscript.

In the latest round of comments, Reviewer 2 has emphasized the crucial need for hardware innovation. They have highlighted what they consider our inadequacy in addressing key challenges that hinder optical neural networks from surpassing their electronic counterparts. While we acknowledge Reviewer 2's concerns, we posit that these observations should not be deemed sufficient grounds for the rejection of our work from publication in NC. Across the preceding two review rounds, we diligently presented a comprehensive and clear demonstration of the innovation and superiority embedded in our work. We are open to a thorough discussion with Reviewer 2 regarding the raised issues.

1. Innovation and Comparative Analysis:

In the previous round of peer review, we extensively compared the hardware aspect of our work with other highly impactful studies. Specifically addressing the raised concerns about limited scalability, we have not only effectively resolved this issue but

also highlighted the unique scalability of our HyArch PIC architecture. Acknowledging the challenge posed by high photonic conversion costs, we recognize that the main bottleneck resides in the electronic domain. Overcoming this bottleneck requires substantial technological advancements, making it a complex challenge at this stage. Regarding storage requirements, the intrinsic difficulty of optical storage has impeded its application in large-scale computing tasks. Although research on photonics-enabled storage is still in its early stages, employing Phase Change Materials (PCM) for optical storage emerges as a promising approach.

Recently, a noteworthy review on optical computing published in Nature Review Physics has captured our attention (McMahon, P.L. The physics of optical computing. Nat Rev Phys (2023)). This comprehensive article systematically discusses ten key challenges that optical computing schemes and architectures may surmount. These challenges include **optical-processor architecture design, applications**, nonlinearity, cascadability, 3D design and manufacturing, energy costs for electronic and optoelectronic components, **scale, robustness, reliability, and fabrication variation**, storage, and pushing towards quantum limits. Our research, as evidenced, effectively addresses the majority of these key challenges and demonstrates innovative breakthroughs in several aspects. The HyArch PIC **innovates the architecture of PIC**, significantly enhancing its functionality to **support more intricate applications**, including advanced artificial intelligence algorithms such as reinforcement learning. Moreover, from a hardware architecture perspective, our hybrid architecture exhibits substantial advantages in terms of **scale, robustness, reliability, and fabrication variation**. In comparison to traditional MZI mesh structures, the HyArch PIC architecture demonstrates greater scalability and notable advantages in robustness and fabrication variation. Our analysis of the energy costs for electronic and optoelectronic components within the entire architecture has led to proposed improvement strategies.

It is essential to highlight two relevant studies published in high-impact journals: "Experimental quantum speed-up in reinforcement learning agents" by Saggio, Valeria, et al. (*Nature* **591**, 229–233 (2021)), and "Experimental photonic quantum memristor" by Spagnolo, Michele, et al. (*Nat. Photon.* **16**, 318–323 (2022)). Despite solely utilizing MZI structures, these works showcase diverse physical and algorithmic applications, earning significant citations and recognition. Thus, criticizing our work based on the notion that: *"the study still falls within the scope of demonstrating an algorithm using existing hardware methods. The techniques employed, including the linear optical Mach-Zehnder interferometer (MZI) network and active optoelectronic control, are mature technologies in this field."* is unjust. MZI networks inherently embody rich physics, and within the constraints of our capabilities, we have pushed

hardware performance to its limits, achieving a paradigm shift and demonstrating record-level encoding dimensions and computational accuracy.

2. ONN vs. Electronic Counterparts: Realizing True Superiority and Future Coexistence Trends:

Reviewer 2 has raised the concern that our work *"fails to address these critical challenges in optical neural networks that hinder optical neural networks from truly surpassing their electronic counterparts."* This specific issue has undergone a detailed discussion in our manuscript, particularly in Supplementary Section 8, where we delve into the computational scalability achievable by HyArch PIC and its corresponding electronic counterparts across various scales and system frequencies.

It is crucial to recognize that achieving genuine computational superiority, often referred to as photonic superiority, demands substantial technological accumulation—an objective currently beyond reach. The prevailing industry viewpoint emphasizes that achieving widespread implementation of photonic computing is a long-term undertaking. Currently, there is an ongoing exploration of specialized algorithms for photonics acceleration. Looking ahead, photonics computing is poised to evolve as a specialized form of acceleration rather than serving as general-purpose hardware, akin to CPUs.

Within the field of optoelectronic computing, an increasing body of work is concentrating on leveraging the strengths of both electronic and photonic domains to achieve hybrid integration, thereby enhancing overall system efficiency at the systemic level, rather than pursuing a unilateral superiority stance. Our research work stands as a compelling illustration of this integrated optoelectronic architecture. Through the integration of multi-channel electrical driving, PIC, and detector co-packaging, we markedly elevate the integration level of the entire optoelectronic computing system. Furthermore, the utilization of Co-Packaging Optics (CPO) in 3D design and manufacturing holds the potential to further amplify the computational power of optoelectronic computing systems, paving the way for a new generation of AI computing architectures.

3. Our Efforts to Tackle the Above Challenges:

Revisiting the pivotal challenges highlighted by Reviewer 2 that impede the progress of photonic computing, we are actively engaged in overcoming these obstacles to enhance the overall performance of optoelectronic computing architectures.

In terms of scalability, beyond the adoption of a suitable computing architecture like our HyArch PIC, a major challenge is posed by the large-scale fabrication of PICs. Transmission losses and device stability significantly influence the quality of the final photonic computing output. To enhance the integration level of photonic integrated computing chips, a stable fabrication platform stands as a prerequisite. After evaluating conventional Process Design Kit (PDK) conditions on widely used open platforms, we conclude that collaborative research and development with fabrication facilities or the independent establishment of a pilot line are imperative steps toward realizing large-scale photonic computing. Realizing ultra-large-scale photonic computing chips relies on the continuous advancement of the parameter conditions for photonic chips. There are already ongoing efforts in this direction.

Addressing the challenge of high photonic conversion costs, we suggest a potential solution by advocating the use of Lithium Niobate on Insulator (LNOI) material in the PIC, as opposed to the conventional Silicon Photonics (SiPh). Leveraging LNOI's robust electro-optic characteristics, we can attain low-loss, high-speed electro-optic conversion, thereby substantially reducing energy consumption related to electro-optic conversion. This approach fully capitalizes on the high bandwidth features of photonic computing.

In conclusion, we express our sincere appreciation to Reviewer 2 for their insightful feedback and valuable suggestions. We look forward to continuous advancements in photonic computing hardware performance in future endeavors, with the ultimate goal of realizing a genuinely high-powered and energy-efficient optoelectronic hybrid computing system. Additionally, we kindly request Reviewer 2 to conduct a comprehensive reassessment of our work, taking into account the current technological landscape and comparative analyses with other studies. While ensuring ample originality, our experimental demonstrations showcase innovative architectures and algorithms, presenting a promising new direction in integrated photonic computing. We remain confident that our work aligns with the high standards of Nature Communications.

Response to the Report of Reviewer #3

Reviewer #3 (Remarks to the Author):

In the first revision, my main concern was that the novelty was not coming through in the manuscript. In the 2nd revision, I am satisfied with the authors' attempt to further clarify the claims and novelty. So, in principle, I would recommend this manuscript for publication. However, all of this discussion is relegated to the discussion section. I would recommend that some of that discussion, especially on the advantages of hybrid architecture (over the standard unitary MZI mesh and coherent linear architectures), be explicitly stated in the introduction.

I thank the authors for considering my suggestions.

We appreciate Reviewer 3's constructive feedback in the final round of review. The insightful suggestion to explicitly incorporate discussions on the advantages of the hybrid architecture in the introduction has been duly considered. Following this recommendation, we have revised the introduction section to emphasize the distinct benefits of the hybrid architecture compared to the standard unitary MZI mesh and coherent linear architectures. This enhancement aims to better highlight the novelty and contributions of our work from the outset. The revised introduction section, incorporating the suggested modifications, is presented below:

“Additionally, the progress of integrated optical computing is impeded by limitations inherent in single architectures, such as Mach-Zehnder interferometers (MZI) mesh and coherent linear architectures, which include restricted scalability and functionality.”

“In this work, we experimentally demonstrate the RL efficiency improvements by using the PIC platform to implement agent-environment interaction. We design a hybrid architecture PIC (HyArch PIC) with remarkable scalability and versatile functionality compared to single integrated optical computing architectures. Co-integrating HyArch PIC with a high-speed FPGA and electrical drivers on the same development board, our optoelectronic computing board achieves high integration and large optimization space. ...”

We are grateful for Reviewer 3's thoughtful input, which has undoubtedly contributed to the overall improvement of our manuscript.